# Fine-Grained Alignment and Noise Refinement for Compositional Text-to-Image Generation

## Abstract

Text-to-image generative models have made significant advancements in recent years; however, accurately capturing intricate details in textual prompts—such as entity missing, attribute binding errors, and incorrect relationships remains a formidable challenge. In response, we present an innovative, training-free method that directly addresses these challenges by incorporating tailored objectives to account for textual constraints. Unlike layout-based approaches that enforce rigid structures and limit diversity, our proposed approach offers a more flexible arrangement of the scene by imposing just the extracted constraints from the text, without any unnecessary additions. These constraints are formulated as losses—entity missing, entity mixing, attribute binding, and spatial relationships—integrated into a unified loss that is applied in the first generation stage. Furthermore, we introduce a feedback-driven system for fine-grained initial noise refinement. This system integrates a verifier that evaluates the generated image, identifies inconsistencies, and provides corrective feedback. Leveraging this feedback, our refinement method first targets the unmet constraints by refining the faulty attention maps caused by initial noise, through the optimization of selective losses associated with these constraints. Subsequently, our unified loss function is reapplied to proceed the second generation phase. Experimental results demonstrate that our method, relying solely on our proposed objective functions, significantly enhances compositionality, achieving a 24% improvement in human evaluation and a 25% gain in spatial relationships. Furthermore, our fine-grained noise refinement proves effective, boosting performance by an average of 3% across all categories of the T2I-CompBench benchmark.

## 1 Introduction

Recent advancements in diffusion-based text-to-image (T2I) models (Rombach et al., 2022; Nichol et al., 2021; Saharia et al., 2022; Ramesh et al., 2022; Balaji et al., 2022) have significantly improved the generation of high-quality and diverse images from textual prompts. However, these models often fail to precisely capture the intended meaning, resulting in inconsistencies between the generated images and the original prompt. Recent studies (Huang et al., 2023a; Hu et al., 2023; Meral et al., 2024; Guo et al., 2024) highlight key failure modes in text-to-image generation, including *entity missing*, *attribute binding errors*, and *incorrect relationships*. In response, training-free approaches have been introduced to tackle the challenge of compositionality. The first approach uses spatial layouts or bounding boxes to guide generation (Li et al., 2023; Zhang et al., 2024b; Wang et al., 2024; Zheng et al., 2023). While these methods enhance spatial coherence by enforcing structured layouts, they often struggle with visual realism and maintaining aesthetic quality (Zhang et al., 2024b; 2025). The second branch utilizes Large Language Models (LLMs) to decompose complex tasks into manageable subtasks (Lian et al., 2023; Yang et al., 2024; Li et al., 2024a; Ye et al., 2024). The effectiveness of these methods heavily depends on the capabilities of the underlying LLMs and the quality of prompt engineering (Yang et al., 2024; Zhang et al., 2025). The third branch focuses on optimizing attention maps (Chefer et al., 2023b; Meral et al., 2024; Li et al., 2024b; Singh & Zheng, 2023; Agarwal et al., 2023; Guo et al., 2024). Although these methods typically address one or two specific failure modes, they fail to comprehensively resolve all of them. For instance, A-STAR (Agarwal et al., 2023), A&E (Chefer et al., 2023b),

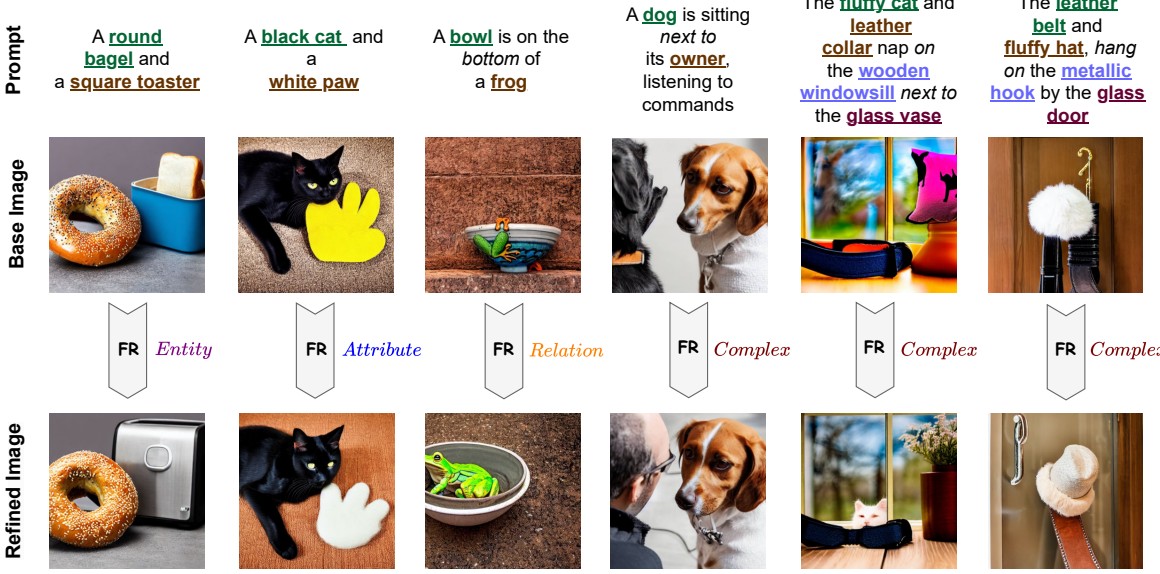

Figure 1: Our proposed fine-grained initial noise refinement (denoted as FR) mitigates various compositional challenges, including entity missing, attribute binding, and spatial relationships.

and INITNO (Guo et al., 2024) primarily tackle entity mixing and missing issues, while Divide-and-Bind (Li et al., 2024b) and CONFORM (Meral et al., 2024) go further by addressing attribute-binding errors as well. Yet, a unified approach that holistically addresses compositionality challenges remains an open problem. We follow the attention-map approach and propose a novel training-free method that defines objectives to arrange the scene based on the entities and their relations mentioned in the prompt. To achieve this, we define four loss functions: (1) *object missing loss*, which encourages the presence of each object in the attention map; (2) *object mixing loss*, which ensures that each object occupies a distinct space in the attention map; (3) *attribute-binding loss*, which guides the attention maps of an entity and its attributes to cover the same area, thereby forcing an entity to have its own attributes; and (4) *spatial relation loss*, which constrains the attention maps of entities to align with the spatial relationships described in the corresponding text components.

Moreover, we take an additional step by designing a feedback-driven system that refines the initial noise of the first generation stage. This system incorporates a verifier that evaluates the generated image and provides feedback on inconsistencies. Our approach leverages feedback signal to identify and correct faulty regions within the initial noise. Specifically, we expand the initial noise into attention maps, refine the faulty ones based on the feedback, by adjusting the noise, thus providing a better starting point for the generation process. Then, the second phase proceeds by applying our loss functions during the generation process. To the best of our knowledge, this form of selective refinement for initial noise has not been explored in previous studies. In contrast to our approach, existing methods such as INITNO (Guo et al., 2024), CoCoNO (Sundaram et al., 2024), and ReNO (Eyring et al., 2024) regenerate the entire noise without distinguishing which parts are faulty. For example, in Fig. 1, given the prompt *"a dog is sitting next to its owner"*, the first generated image suffers from an *entity missing* issue, where the "owner" is not correctly represented. The verifier identifies this inconsistency and provides feedback. Consequently, our method automatically activates the entity missing loss and applies it to the owner's attention map in the initial noise, and then proceeds with the generation process guided by our losses. Also, this form of initial noise refinement can be generalized to handle multiple objects and failure modes simultaneously. For instance, in Fig. 1, given the prompt *"The leather belt and fluffy hat hang on the metallic hook by the glass door"*, the initial generated image not only omits the *hat* but also fails to render the *door* with a glass-like appearance. Our approach leverages these feedback signals to correct both issues. while preserving correctly generated elements in the

noise, such as the belt. We use DA-Score (Singh & Zheng, 2023) as our primary verifier, enhancing its fine-grained question extraction with additional coarse-grained questions. These questions are grouped by failure mode, allowing for precise error detection. The primary contributions of our work are outlined as follows:

- We propose an interpretable set of objective functions that are designed to complement each other and work collaboratively to arrange the image scene.

- A fine-grained initial noise refinement framework is designed to mitigate the generation problems of multiple entities by only incorporating one additional generation stage.

- We conduct comprehensive experiments on 9 baselines, 2 datasets, and 5 verifiers to validate our claims. Our spatial loss delivers a substantial 25% improvement in spatial relationships, and our fine-grained noise refinement surpasses all baselines, achieving a 3% average performance gain across all categories of the T2I-CompBench benchmark .

## 2 Related Works

Before the emergence of diffusion-based models, various research directions aimed to achieve realistic and high-quality image generation, both conditional and unconditional, through approaches like generative adversarial networks (GANs) (Kang et al., 2023; Xu et al., 2018; Ye et al., 2021; Zhang et al., 2021; Zhu et al., 2019; Zhang et al., 2017) and autoregressive models (Chang et al., 2023; Ding et al., 2021; Ramesh et al., 2021; Yu et al., 2022). Recently, diffusion-based models have shown remarkable performance in conditional text-to-image generation (Rombach et al., 2022; Nichol et al., 2021; Saharia et al., 2022; Ramesh et al., 2022; Balaji et al., 2022). While having a good performance, they often have alignment problems when generating images. We discuss previous studies that attempt to alleviate alignment problems under three categories.

**Training-Based Compositional Improvement.** A group of studies have focused on overcoming compositional challenges in text-conditional diffusion-based models to improve alignment between text prompts and generated images. Some approaches address this through training-based methods (Li et al., 2023; Yang et al., 2023; Mou et al., 2024; Zhang et al., 2023; Huang et al., 2023b; Zhang et al., 2025; Eyring et al., 2024). For instance, T2I-Adapter (Mou et al., 2024) and ControlNet (Zhang et al., 2023) target the control of semantic structures by specifying high-level features. ReCo (Yang et al., 2023) refines spatial awareness using adapters on top of the diffusion models, while GLIGEN (Li et al., 2023) integrates grounding information into newly trainable layers. Composer (Huang et al., 2023b) decomposes images into key factors, training a diffusion model with these factors as conditions to recompose the input. Alternatively, Itercomp (Zhang et al., 2025) utilizes iterative feedback learning to enhance the compositional generation, and ReNo (Eyring et al., 2024) adopts reward-based noise optimization for improved alignment. While being effective, adopting these methods comes with the cost of additional training time and resources.

**Training-Free Compositional Improvement.** To mitigate compositional problems, some methods leverage training-free methods optimizing latent or attention maps during inference (Liu et al., 2022; Feng et al., 2022; Chefer et al., 2023a; Agarwal et al., 2023; Meral et al., 2024; Guo et al., 2024; Marioriyad et al., 2024; Ma et al., 2025; Li et al., 2025; Yu & Gao, 2025). For instance, Composable Diffusion (Liu et al., 2022) computes separate denoising latent for each statement and combines them with a score function, or Structure Diffusion (Feng et al., 2022) manipulates the cross-attention maps guided by hierarchically extracted structures from text. Attend-and-Excite proposed a novel loss based on cross-attention maps to reduce missing objects. Similarly, A-STAR (Agarwal et al., 2023) optimizes latent representations by employing segregation and retention losses to minimize cross-attention overlap between different concepts, leading the objects not to be overlooked while preserving cross-attention information across all concepts. Also, Divide-and-Bind (Li et al., 2024b) addresses the problem of entity missing by leveraging an attendance objective. Additionally, it enhances attribute association through the incorporation of a binding objective. Similarly, CONFORM (Meral et al., 2024) addresses entity missing and attribute binding issues by employing a contrastive objective. This approach brings each entity and its attribute's attention map closer together while disentangling the attention maps of different entities and their respective attributes. In another approach (Guo et al.,

2024), the initial noise is optimized to ensure it lies within a valid space for generating images aligned with the input prompt. Another approach involves utilizing evaluation feedback to refine the image after its generation. The Evaluate-and-Refine method (Singh & Zheng, 2023) employs iterative VQA feedback to adjust the weighting of CLIP's phrase embeddings. Additionally, it integrates the Attend-and-Excite method to address the issue of entity missing effectively. However, these approaches are restricted in handling compositional problems such as spatial relationships, entity missing, and attribute binding in an integrated manner while utilizing textual constraints simultaneously enables us to arrange the scene more effectively.

**Leveraging LLMs for Accurate Alignment in Text-to-Image Models.** Recent works have leveraged layout-based approaches to enhance compositional alignment in images generated by text-to-image models, using spatial layouts or bounding boxes (Lian et al., 2023; Zhang et al., 2024b; Li et al., 2023; Wang et al., 2024; Zheng et al., 2023). Although these methods enhance spatial awareness, they face challenges in achieving realistic image generation, particularly in creating non-spatial relationships and maintaining aesthetic quality (Zhang et al., 2024b; 2025). Other studies utilizing LLMs aim to break down the generation of an aligned image by converting the prompt into simpler subtasks, with the planning of these subtasks throughout the generation process (Lian et al., 2023; Ye et al., 2024; Li et al., 2024a; Yang et al., 2024; Park et al., 2025). However, these models often struggle to produce accurate results due to the inherent complexity of LLM outputs (Yang et al., 2024; Zhang et al., 2025). Additionally, since these methods also extract spatial layouts or bounding boxes, they encounter challenges similar to those faced by layout-based methods.

## 3 Preliminaries

**Stable Diffusion.** We apply our alignment and refinement method on the Stable Diffusion model (Rombach et al., 2022). In SD, an encoder $\mathcal{E}$ is trained to map an image $x \in \mathcal{X}$ to a latent representation $z = \mathcal{E}(x)$. The latent code $z$ is then given to a decoder model to reconstruct the input image such that $\mathcal{D}(\mathcal{E}(x)) \approx x$. After training the autoencoder, a denoising diffusion probabilistic model (DDPM) (Ho et al., 2020) is trained over the latent space of the autoencoder. The UNet (Ronneberger et al., 2015) denoising model $\epsilon_\theta$ learns to denoise an input latent $z_t$ at each timestep $t$ (where $z_t$ results from adding $\epsilon$ noise gradually over $t$ timesteps to the original latent $z_0$ during training). The denoising objective which intends to learn $\epsilon_\theta$ in order to predict the noise $\epsilon$ is given by:

$$\mathcal{L} = \mathbb{E}_{z, \mathcal{P}, \epsilon \sim \mathcal{N}(0, I), t} \left[ ||\epsilon - \epsilon_\theta(z_t, t, L(\mathcal{P}))||_2^2 \right] \tag{1}$$

The denoising process of Stable Diffusion is conditioned on the embedding of text information $L(\mathcal{P})$. In Stable Diffusion, this embedding is the output of the CLIP (Radford et al., 2021) text encoder. At inference time, a random latent $z_t$ is sampled from $\mathcal{N}(0, I)$, and the trained UNet denoising model $\epsilon_\theta$ outputs a denoised latent $z_0$. We obtain the reconstructed image by feeding $z_0$ to the decoder $\mathcal{D}$.

**Stable Diffusion Cross-Attention.** The intermediate image outputs of denoiser UNets are conditioned on text-encoder embeddings through the cross-attention mechanism. The key and values, $K$ and $V$, are projections of $L(\mathcal{P})$, and the queries, $Q$, are derived from the intermediate representation of UNet. The cross-attention map at time $t$ is defined as $A^t = \text{Softmax}\left(\frac{QK^T}{\sqrt{d}}\right)$ where $d$ is projection space dimension.

The cross-attention map has the shape of $h \times w \times l$ where $h$ and $w$ are feature map resolution and $l$ is the number of text tokens. We set feature map resolution to $16 \times 16$ as it empirically contains the most semantically meaningful attention maps (Hertz et al., 2022).

## 4 Definition

We define key terms and functions that will be used in the following sections. let $A_i^t$ represent the cross-attention maps of token $i$ at time step $t$, normalized using the max-min method Guo & Lin (2023). We define the following token sets:

- $G_{\text{entity}}$: The set of entity token indices.

- $G_{\text{attribute}}$: The set of all entity-attribute token index pairs.

- $G_{\text{relation}}$: The set of all tuples $(e_1, r, e_2)$, where token $e_1$ has a spatial relationship $r$ with token $e_2$.

- $\mathbb{I}_x(r)$: An indicator function that determines whether the spatial relation $r$ is defined along the $x$-axis. e.g. $\mathbb{I}_x(\text{left})$ is 1 but $\mathbb{I}_x(\text{top})$ is 0. The function $\mathbb{I}_y(r)$ is defined analogously for the $y$-axis, following the same principle.

- $\text{dir}(r)$: Indicates whether the spatial relation $r$ points in the same direction as its underlying axis. For instance, $\text{dir}(\text{right})$ is 1 since the positive side of $x$-axis is on the right side, whereas $\text{dir}(\text{left})$ is $-1$ based on the same principle.

We used Large Language Models specifically, GPT-4o for extracting $G_{\text{entity}}$, $G_{\text{attribute}}$, and $G_{\text{relation}}$, from questions. More details are available in Appendix D (under the Prompt Decomposition section). The Intersection-over-Union (IoU) of attention maps evaluates the overlap between attention maps of two tokens. It is defined as:

$$IoU\left(A_m^t, A_n^t\right) = \frac{\sum_{ij}\left([A_m^t]_{ij} \times [A_n^t]_{ij}\right)}{\sum_{ij}\left([A_m^t]_{ij} + [A_n^t]_{ij}\right)}. \tag{2}$$

The center of mass of an attention map $A$ along the x-axis is defined as $E_x(A) = \sum_i i \sum_j A_{i,j}$. Same goes for $E_y(A)$.

## 5 Method

We propose a novel training-free method that defines four objectives for arranging a scene based on the entities and their relationships mentioned in the prompt. We collectively refer to these four objectives as EAR loss (**E**ntity-**A**ttribute-**R**elation loss), which comprises three components: Entity loss (object missing, object mixing), Attribute loss (attribute binding), and Relation loss (spatial relation). In addition, we sometimes observe that applying our EAR loss to the initial noise is not sufficient to generate a well-aligned image. To address this issue, we propose an innovative fine-grained initial noise refinement method that optimizes the initial noise based on feedback from the verifier in the first generation stage.

### 5.1 Fine-grained Alignment by EAR Loss

Our method iteratively modifies latent space during specific inference steps by introducing objective functions, each aimed at a particular challenge. These objective functions rely on cross-attention maps to locate the occurrence of each entity and attribute within the image to arrange the image scene properly.

**Entity**: To address the entity problem, we introduce two loss functions: entity mixing loss and entity missing loss. For entity mixing, we minimize the overlap between attention maps of each entity pair using the IoU, formulated as:

$$\mathcal{L}_{\text{mixing}} = \sum_{e_1, e_2 \in G_{entity}} IoU\left(A_{e_1}^t, A_{e_2}^t\right). \tag{3}$$

For entity missing, we do not strongly favor the attention map for entities to have minimal overlap with others, but we also ensure that the amount of space exclusively allocated to each entity is maximized. This is formulated as:

$$\mathcal{L}_{\text{missing}} = -\sum_{e_1 \in G_{entity}} \frac{\sum_{e_2 \in G_{entity}} max\left(A_{e_1}^t - A_{e_2}^t, 0\right)}{n-1}, \tag{4}$$

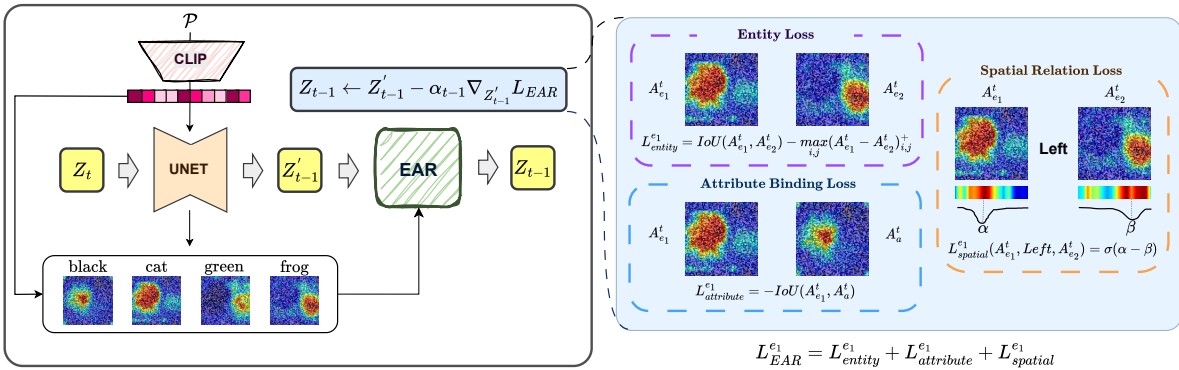

Figure 2: **Illustration of EAR Losses.** Given the prompt *A black cat is on the left of a green frog*, attention maps are extracted at each denoising step to compute our proposed losses; (1) Entity Loss: Reduces entity mixing and missing by minimizing the overlap between attention maps of each entity pairs while maximizing the amount of space exclusively allocated to each entity. (2) Attribute Binding Loss: Brings attention maps of attributes closer to their corresponding entities. (3) Spatial Relation Loss: Shifts the distributions of the two entities' attention maps toward their correct relative positions.

where $n = |G_{entity}|$ is total number of entities. This objective measures the amount of space exclusively allocated to an entity and aims to maximize it by enhancing the distinction between its attention map and those of other entities. We emphasize the necessity of using both losses, as their combination not only prevents entity mixing but also reinforces the generation of each entity.

**Attribute**: Our attribute binding's objective function aims to bring the attention map of each attribute and its corresponding entity closer together to ensure they cover the same area. We maximize their overlap as follows:

$$\mathcal{L}_{attr} = - \sum_{(ent,attr) \in G_{attribute}} IoU\big(A_{ent}^t, A_{attr}^t\big), \tag{5}$$

where $G_{attribute}$ is the set of all entity-attribute pairs.

**Relation**: In this study, we focus on addressing spatial relations. To achieve this, we first compute the center of mass for each entity's attention map along each axis. Next, we determine the relation type: vertical or horizontal. Depending on the relationship type and the centers of mass, we optimize the latent space to shift the distributions of the two entities' attention maps toward their correct relative positions. Our optimization is described as:

$$\mathcal{L}_{spatial} = \sum_{(e_1,r,e_2) \in G_{relations}} \Big[ \mathbb{I}_x(r) \cdot \sigma\big(dir(r) \cdot \big(E_x(A_{e_2}^t) - E_x(A_{e_1}^t)\big)\big) + \mathbb{I}_y(r) \cdot \sigma\big(dir(r) \cdot \big(E_y(A_{e_2}^t) - E_y(A_{e_1}^t)\big)\big) \Big] \tag{6}$$

where $\mathbb{I}(.)$ and $dir(.)$ represent the indicator and direction functions, respectively. Both are formally defined in 4. To illustrate the above formula, in Fig. 2, we consider the triple (cat, left, frog). The spatial loss formula for this case is expressed as:

$$\mathcal{L}_{spatial} = \sigma(E_x(A_{cat}^t) - E_x(A_{frog}^t)) \tag{7}$$

**EAR**: Finally, the overall EAR loss, taking into account various constraints related to the scene, is defined as:

$$\mathcal{L}_{EAR} = \underbrace{(\mathcal{L}_{missing} + \mathcal{L}_{mixing})}_{\mathcal{L}_{entity}} + \mathcal{L}_{attr} + \mathcal{L}_{spatial} \tag{8}$$

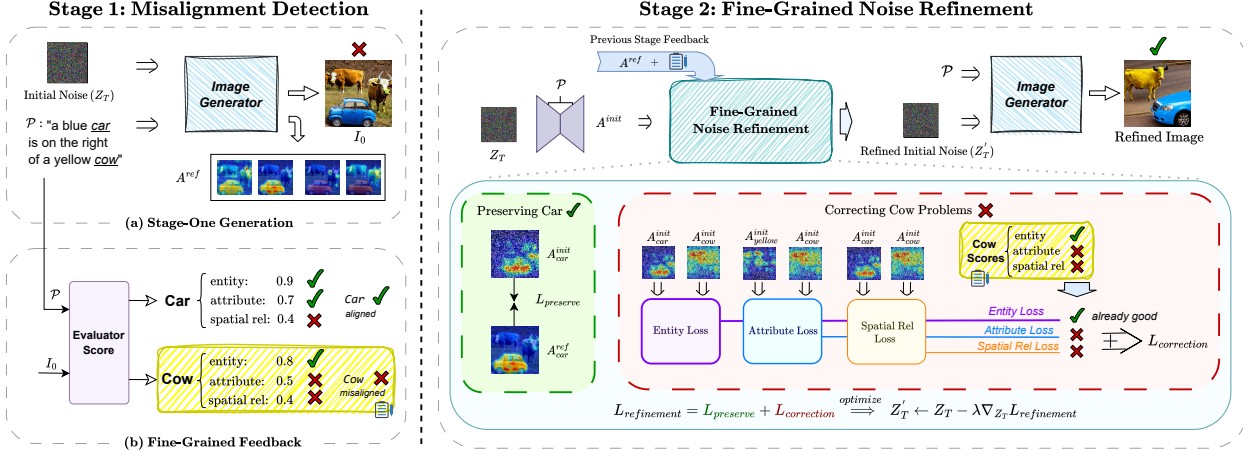

Figure 3: **Illustration of fine-grained initial noise refinement method.** (1) Misalignment Detection: an image is generated and assessed by the verifier, which scores the alignment of entities, attributes, and spatial relationships with the prompt and provides detailed feedback. (2) Fine-Grained Noise Refinement: The initial noise is refined by sequentially correcting each misaligned entity through targeted adjustments, leveraging a weighted sum of specialized loss functions designed to address the issues identified through feedback. An additional loss function is introduced to preserve the quality of already aligned entities by refining the attention maps of the initial noise. Once all misaligned entities are corrected, the remaining generation process proceeds similarly to the first stage.

**Latent Update**: During the first 25 steps of the diffusion process, the latent space at timestep $t$, denoted as $Z_t$, is denoised using the U-Net (Ronneberger et al., 2015), yielding $Z'_{t-1}$. It is then refined using the following update rule:

$$Z_{t-1} \leftarrow Z'_{t-1} - \alpha_{t-1} \nabla_{Z'_{t-1}} \mathcal{L}_{\text{EAR}} \tag{9}$$

where $\alpha_t$ is the step size of the gradient update. The complete process of updating the latent space is illustrated in Fig. 2, and the outlined EAR loss generation procedure is encapsulated in Algorithm 1.

## 5.2 Fine-grained Initial Noise Refinement

We propose a feedback-driven system for fine-grained initial noise refinement, enabling precise refinement by selectively correcting only the impaired components. In the first stage of our system, we generate an image using our EAR loss. We then employ a verifier to provide fine-grained feedback. This entire stage is referred to as misalignment detection. In the second stage, we utilize the obtained feedback. Our refinement method first addresses the unmet constraints by correcting the faulty attention maps caused by the initial noise, optimizing selective losses related to these constraints. Following this, our unified loss function is reapplied to initiate the second generation phase. This stage is known as fine-grained noise refinement. The complete workflow of these two stages is illustrated in Fig. 3.

**Misalignment Detection**: In this stage, we assess the quality of the first generated image and identify potential flaws. To achieve this, we select a verifier, such as DA-Score (Singh & Zheng, 2023), and adapt it to provide fine-grained feedback on specific types of issues present in the image. An example of this misalignment detection process is illustrated on the left side of Fig. 3, Further details on misalignment detection and verifier adaptation are provided in Supp. B.

**Fine-Grained Noise Refinement**: After receiving feedback from the misalignment detection stage, we apply a fine-grained method to improve the initial noise. We identify problematic attention maps in the noise as *faulty* entities and attempt to correct them while preserving the *proper* entities. During refinement, we iterate over the *faulty* set, picking a *faulty* entity at each step of refinement and select proper losses in response to the type of mistakes associated with it. After correcting an entity, we move it to the *proper* set. We formulate the correction loss as follows:

$$\mathcal{L}_{correction} = \mathbb{I}_{\text{entity}}\mathcal{L}_{\text{entity}}^{f} + \mathbb{I}_{\text{attribute}}\mathcal{L}_{\text{attribute}}^{f} + \mathbb{I}_{\text{spatial}}\mathcal{L}_{\text{spatial}}^{f}, \tag{10}$$

where $f$ represents the selected *faulty* entity, and $\mathcal{L}^{f}$ denotes the loss calculation specific to this entity. The indicator function is defined as $\mathbb{I}_{\text{mistake}} = \mathbb{I}(\text{score}_{\text{mistake}} \geq \lambda)$, where $\lambda$ is a hyperparameter.

Simultaneously, we apply a preservation loss to the set of *proper* entities, denoted as $G_{\text{proper}}$, to ensure their quality is retained. Let $\{A^{\text{init}}\}_{i=1}^{n}$ represent the cross-attention maps of these entities after the current denoising step, and let $\{A^{\text{ref}}\}_{i=1}^{n}$ denote their corresponding target reference attention maps. The definition of the reference attention map $A^{\text{ref}}$ depends on the entity's history. For entities that were proper in the original input, we use the cross-attention maps from the previous stage as the reference. However, for entities that were originally faulty but fixed in preceding steps (the "added" proper entities), we adopt their modified maps as the reference. We formulate the preservation loss as:

$$\mathcal{L}_{preservation} = - \sum_{ent \in G_{\text{proper}}} IoU(A_{ent}^{init}, A_{ent}^{ref}). \tag{11}$$

Overall, the refinement loss function is expressed as:

$$\mathcal{L}_{refinement} = \mathcal{L}_{correction} + \mathcal{L}_{preservation} \tag{12}$$

**Initial Noise Update**: The initial noise is refined according to the following formula:

$$Z'_{T} \leftarrow Z_{T} - \alpha \nabla_{Z_{T}} \mathcal{L}_{refinement} \tag{13}$$

where $\alpha$ represents the step size for the initial noise gradient update. A visualization of the fine-grained noise refinement process is shown on the right side of Fig. 3, and the pseudocode for the fine-grained initial noise refinement is described in Algorithm 2.

---

**Algorithm 1** EAR Generation

---

**Input:** Prompt ($\mathcal{P}$), Number of Denoising Steps ($T$), EAR loss ($L_{\text{EAR}}$), stopping step ($t_{max}$), Entity Set ($G_{\text{entity}}$), Attribute Set ($G_{\text{attribute}}$), Spatial Relation Set ($G_{\text{relation}}$), Stable Diffusion Model ($SD$), Decoder ($D$)
**Output:** Image ($\mathcal{I}$)

1: **for** $t$ in $[T...1]$ **do**
2:     $Z'_{t-1}, A^{t} \leftarrow SD(Z_{t}, \mathcal{P})$
3:     **if** $t > t_{max}$ **then**
4:         $L_{\text{EAR}} \leftarrow$ Compute $L_{\text{EAR}}$ using $G_{\text{entity}}$,
5:             $G_{\text{attribute}}, G_{\text{relation}}$ and $A^{t}$          ▷ *Eq. 8*
6:         $Z_{t-1} \leftarrow Z'_{t-1} - \alpha_{t-1}\nabla_{Z'_{t-1}}L_{\text{EAR}}$          ▷ *Eq. 9*
7:     **else**
8:         $Z_{t-1} \leftarrow Z'_{t-1}$
9:     **end if**
10: **end for**
11: $\mathcal{I} \leftarrow D(Z_{0})$
12: **return** $\mathcal{I}$

---

# 6 Results

## 6.1 Experimental Settings

**Datasets.** The evaluation is performed on T2I-CompBench (Huang et al., 2023a) and HRS (Bakr et al., 2023). T2I-CompBench assesses complex compositional generation, while HRS is used to evaluate our

---

**Algorithm 2** Fine-Grained Initial Noise Refinement

---

**Input:** Prompt ($\mathcal{P}$), Number of Denoising Steps ($T$), EAR Generation Model ($G$), Initial Noise Learning Rate ($\alpha$), Verifier ($V$), Stable Diffusion Model (SD)
**Output:** Image ($\mathcal{I}$)

1:  **Stage 1: Misalignment Detection**
2:  $Z_T \leftarrow$ Initial Noise
3:  $\mathcal{I}_0, \{A^{\text{ref}}\} \leftarrow G(Z_T, \mathcal{P})$                         ▷ *Initialize references from original generation*
4:  faulty_entities, proper_entities $\leftarrow$ V($\mathcal{I}_0, \mathcal{P}$)                              ▷ *Fig. 3*
5:  **Stage 2: Fine-grained Noise Refinement**
6:  **while** len(faulty_entities) $> 0$ **do**
7:     $A^{\text{init}} \leftarrow \text{SD}(Z_T, \mathcal{P})$                           ▷ *Re-compute current maps from $Z_T$*
8:     $f \leftarrow$ faulty_entities.pop()
9:     $L_{\text{correction}} \leftarrow$ Compute for $f$ using $A^{\text{init}}$                    ▷ *Eq. 10*
10:    $L_{\text{preservation}} \leftarrow$ Compute from proper_entities, $A^{\text{init}}, A^{\text{ref}}$       ▷ *Eq. 11*
11:    $L_{\text{refinement}} \leftarrow L_{\text{preservation}} + L_{\text{correction}}$                  ▷ *Eq. 12*
12:    $Z_T \leftarrow Z_T - \alpha \nabla_{Z_T} L_{\text{refinement}}$                     ▷ *Eq. 13*
13:    proper_entities.add($f$)
14:    $A_f^{\text{ref}} \leftarrow A_f^{\text{init}}$                        ▷ *Update reference for the newly fixed entity*
15:  **end while**
16:  $\mathcal{I} \leftarrow G(Z_T, \mathcal{P})$
17:  **return**  $\mathcal{I}$

---

model's capability in multi-entity generation, we use the HRS dataset's color 3-entity composition category. More information on the datasets is available in the Supp. C.1.

**Baselines.** We compare ourselves with four categories of methods: (1) base models including stable diffusion v1.4, v2 (Rombach et al., 2022) and XL (Podell et al., 2023); (2) approaches focused on latent space optimization such as A&E (Chefer et al., 2023b), CONFORM (Meral et al., 2024), D&B (Li et al., 2024b) and A-STAR (Agarwal et al., 2023); (3) Evaluate&Refine (Singh & Zheng, 2023), which utilizes Visual Question Answering (VQA) feedback; and (4) INITNO (Guo et al., 2024), which iteratively alters the initial noise. In experiments, all baselines, excluding the Stable Diffusion family, are based on Stable Diffusion v1.4.

## 6.2  Quantitative Results

We quantitatively evaluate our approach using multiple verifiers. Table 1 and 2 present the performance of all methods based on the DA-Score (Singh & Zheng, 2023) and TIFA-Score (Hu et al., 2023), respectively. Although some of these models generate images iteratively, our one-stage generation method outperformed the others in most categories. Table 3 presents our results on the HRS dataset, focusing on three-entity prompts from the color category. This experiment highlights the effectiveness of our method in handling multiple entities. Our fine-grained noise refinement significantly enhances the results, achieving improvements of up to 7% on the DA-Score and 4% on the Tifa-Score, respectively. This underscores the effectiveness of our fine-grained noise refinement method in handling multi-object prompts. To ensure a reliable evaluation of generative models on spatial relationships, we employ the VISOR Score (Gokhale et al., 2022), which precisely evaluates spatial relationships. To perform this experiment, we require a larger set of prompts. Therefore, we arbitrarily select 150 spatial prompts from the T2I-CompBench (Huang et al., 2023a) training dataset. As shown in Figure 4(a), integrating our spatial loss with other losses resulted in a substantial 25% boost in the VISOR Score, outperforming other baselines. Additional quantitative results are provided in the Supp. C.4

**User Study.** We conducted a user study with 15 volunteers to evaluate image-text alignment, comparing our method against three baseline models: CONFORM (Meral et al., 2024), INITNO (Guo et al., 2024), and Evaluate&Refine (Singh & Zheng, 2023). Using 25 prompts from T2I-CompBench, volunteers selected the

Table 1: **Performance Comparison**. Evaluation of various models on the DA-Score metric across different categories using the T2I-CompBench benchmark. The term "iter" in this table indicates the number of times the image is generated using that specific method.

| Model | Shape | Color | Texture | Relation | | Complex | Average |
|---|---|---|---|---|---|---|---|
| | | | | Spatial | Non-spatial | | |
| Stable v1.4 | 0.60 | 0.67 | 0.64 | 0.69 | 0.78 | 0.61 | 0.66 |
| Stable v2 | 0.62 | 0.64 | 0.71 | 0.70 | 0.81 | 0.69 | 0.69 |
| Stable-XL | 0.68 | 0.77 | 0.75 | 0.76 | 0.80 | 0.69 | 0.74 |
| A-STAR | 0.61 | 0.74 | 0.74 | 0.68 | 0.79 | 0.65 | 0.70 |
| A&E | 0.63 | 0.75 | 0.79 | 0.76 | 0.77 | 0.70 | 0.73 |
| D&B | 0.66 | 0.77 | 0.73 | 0.75 | 0.77 | 0.68 | 0.73 |
| Evaluate&Refine (3 iters) | 0.66 | 0.72 | 0.76 | 0.70 | 0.80 | 0.67 | 0.72 |
| CONFORM | **0.70** | 0.78 | 0.78 | 0.76 | 0.76 | 0.68 | 0.74 |
| INITNO (5 iters) | 0.67 | 0.73 | 0.81 | 0.76 | 0.77 | 0.66 | 0.73 |
| Ours | 0.65 | 0.78 | 0.79 | 0.78 | 0.78 | 0.71 | 0.75 |
| Ours + FR (2 iters) | **0.70** | **0.82** | **0.82** | **0.80** | **0.82** | **0.74** | **0.78** |

Table 2: **Performance Comparison**. Evaluation of various models on the Tifa-Score metric across different categories using the T2I-CompBench benchmark. The term "iter" in this table indicates the number of times the image is generated using that specific method.

| Model | Shape | Color | Texture | Relation | | Complex | Average |
|---|---|---|---|---|---|---|---|
| | | | | Spatial | Non-spatial | | |
| Stable v1.4 | 0.62 | 0.77 | 0.74 | 0.73 | 0.86 | 0.76 | 0.75 |
| Stable v2 | 0.66 | 0.75 | 0.79 | 0.76 | 0.87 | 0.81 | 0.77 |
| Stable-XL | 0.69 | 0.86 | 0.82 | 0.82 | 0.87 | 0.77 | 0.81 |
| A-STAR | 0.63 | 0.84 | 0.82 | 0.73 | 0.84 | 0.78 | 0.77 |
| A&E | 0.69 | 0.84 | 0.83 | 0.81 | 0.84 | 0.79 | 0.80 |
| D&B | 0.70 | 0.85 | 0.84 | 0.79 | 0.84 | 0.81 | 0.81 |
| Evaluate&Refine (3 iters) | 0.62 | 0.76 | 0.79 | 0.75 | 0.86 | 0.80 | 0.76 |
| CONFORM | **0.71** | 0.87 | 0.83 | 0.83 | 0.86 | 0.81 | 0.82 |
| INITNO (5 iters) | 0.68 | 0.82 | **0.88** | 0.80 | 0.86 | 0.79 | 0.81 |
| Ours | **0.71** | 0.86 | 0.83 | 0.86 | 0.87 | 0.82 | 0.83 |
| Ours + FR (2 iters) | 0.70 | **0.88** | 0.85 | **0.89** | **0.89** | **0.84** | **0.84** |

best-matching image per prompt. Our method outperformed the baselines by 24%, as shown in Figure 4(b). More detail is reported in Supp. C.2.2

**Loss functions Ablation Study** In this ablation study, we evaluate the individual impact of our loss functions. As shown in Table 4, both entity missing and mixing losses contribute significantly to performance, with the omission of either resulting in a decline, highlighting their complementary effect. Conversely, keeping only the *entity missing* and *mixing* losses while omitting *attribute binding* and *spatial relationship* losses reduces performance on prompts involving attributes or spatial relations. This underscores the necessity of incorporating these additional losses alongside entity losses to achieve robust performance.

**Inference-time and Diversity Comparison** Table 5 presents a comparison of inference time and FID (Heusel et al., 2017) scores across various baselines. We randomly selected 50 sample prompts from T2I-Combench (Huang et al., 2023a) to evaluate and benchmark the performance of our method against existing approaches. Our one stage method achieves an average inference time of $11.11 \pm 0.15$ seconds, closely matching SDv1 ($11.10 \pm 0.13$ seconds) while significantly outperforming computationally heavier methods such as DA-Score (23.26s) and INITNO (32.56s). Moreover, fine-grained initial noise refinement increases the inference time to $22.34 \pm 0.50$ seconds but improves the FID score to 190.47, demonstrating a trade-off between efficiency and generation quality. Compared to other baselines, our method strikes a balance between fast inference and competitive FID performance. Additional comparisons on diversity and aesthetics can be found in Supp. C.3.2.

Table 3: **Performance Comparison**. Evaluation of various models using the HRS benchmark on the DA-Score and Tifa-Score metrics. The term "iter" in this table indicates the number of times the image is generated using that specific method.

| Model | DA-Score | Tifa-Score |
|---|---|---|
| Stable v1.4 | 0.43 | 0.45 |
| Stable v2 | 0.49 | 0.53 |
| A-STAR | 0.57 | 0.61 |
| A&E | 0.65 | 0.64 |
| D&B | 0.60 | 0.60 |
| Evaluate&Refine (3 iters) | 0.39 | 0.47 |
| CONFORM | 0.64 | 0.66 |
| INITNO (5 iters) | 0.66 | 0.69 |
| Ours | 0.67 | 0.70 |
| Ours + FR (2 iters) | **0.74** | **0.74** |

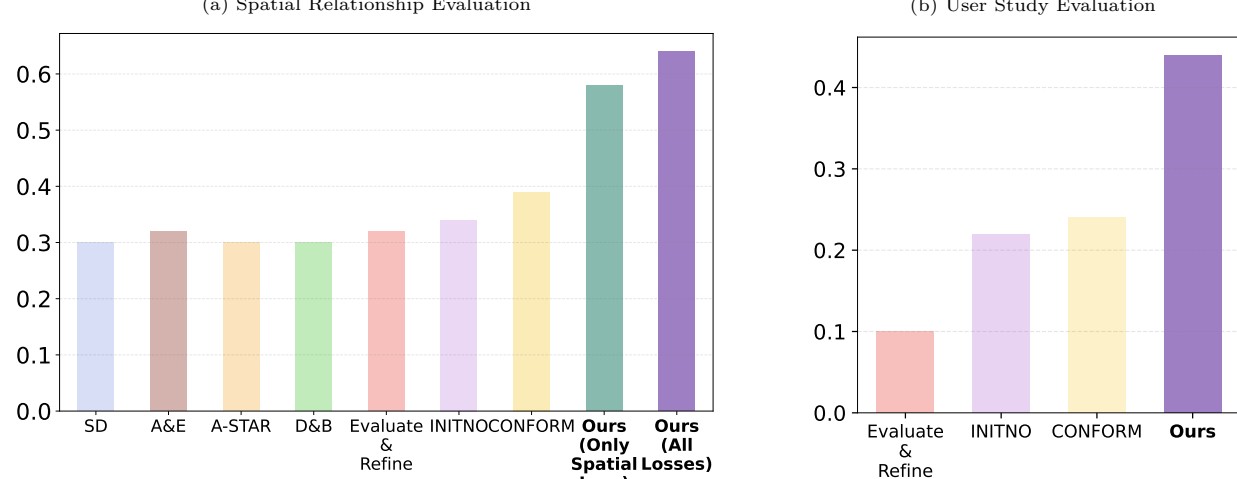

Figure 4: (a) Evaluation of various models on spatial relationship by the VISOR Score using 150 randomly selected prompts from the T2I-Compbench dataset. (b) Evaluation of various models by User study results from 15 participants.

Table 4: Ablation study on different losses. The evaluation is conducted on the DA-Score metric across different categories using the T2I-CompBench benchmark.

| Loss Term | | | | Compositional Categories | | | | | | |
|---|---|---|---|---|---|---|---|---|---|---|
| Entity Mixing | Entity Missing | Attribute Binding | Spatial Relationship | Shape | Color | Texture | Relation | | Complex | **Average** |
| | | | | | | | Spatial | Non-spatial | | |
| ✓ | | | | 0.63 | 0.72 | 0.74 | 0.73 | 0.78 | 0.69 | 0.71 |
| | ✓ | | | **0.66** | 0.75 | 0.76 | 0.73 | 0.77 | 0.69 | 0.73 |
| ✓ | ✓ | | | 0.65 | 0.77 | 0.77 | 0.74 | **0.79** | 0.70 | 0.74 |
| | | ✓ | | 0.56 | 0.65 | 0.66 | 0.67 | **0.79** | 0.62 | 0.66 |
| ✓ | ✓ | ✓ | | 0.65 | **0.78** | **0.79** | 0.75 | 0.78 | **0.71** | 0.74 |
| ✓ | ✓ | ✓ | ✓ | 0.65 | **0.78** | **0.79** | **0.78** | 0.78 | **0.71** | **0.75** |

**Distinction from Test-Time Sampling Methods** We emphasize that our method is different from Test-Time Sampling (TTS) approaches (Ma et al., 2025). While TTS methods typically perform multiple forward passes to select an optimal initial noise for generation, our approach follows a refinement-based

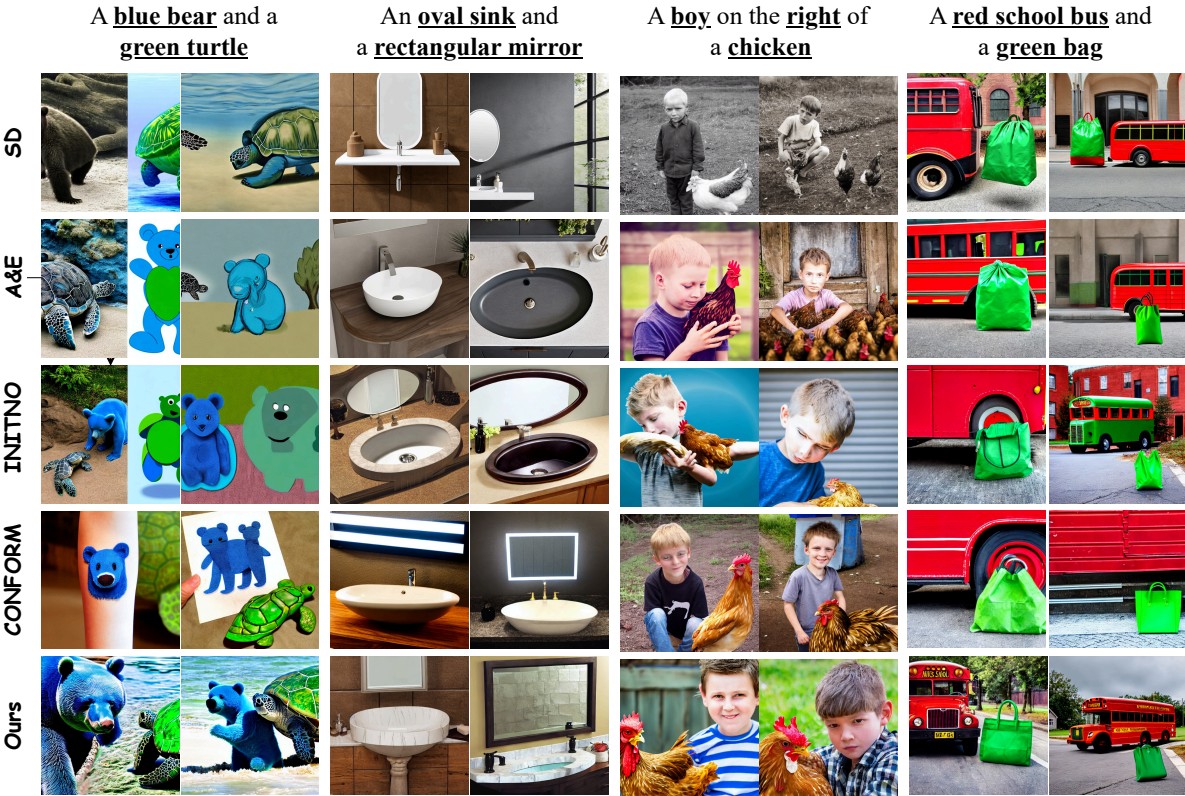

Figure 5: Qualitative Results. We generated two images with distinct random seeds for each of the four example text prompts to enable a qualitative comparison between our method and other models. As shown above, our proposed method produces images that are more realistic and closely aligned with the text prompts. SD refers to Stable Diffusion v1.4.

Table 5: Evaluation of inference time across various models on a set of 50 samples, along with their corresponding FID scores.

| Model | Mean ± Std (Seconds) | FID |
|---|---|---|
| Stable v1.4 | $11.10 \pm 0.13$ | 188.21 |
| A-STAR | $12.49 \pm 0.16$ | 194.31 |
| CONFORM | $21.06 \pm 1.22$ | 193.17 |
| A&E | $16.64 \pm 6.72$ | 190.37 |
| D&B | $17.85 \pm 4.14$ | 194.45 |
| EVALUATION&REFINE (3 iters) | $23.26 \pm 8.55$ | 193.23 |
| InitNO (5 iters) | $32.56 \pm 13.32$ | 190.19 |
| Ours | $11.11 \pm 0.15$ | 192.38 |
| Ours + FR (2 iters) | $22.34 \pm 0.50$ | 190.47 |

strategy. Specifically, we start from a single randomly sampled noise and apply up to two refinement stages to improve it as a better starting point for compositional generation. This makes our task inherently more challenging than standard TTS methods, as not all initial noise samples can be effectively refined. To ensure

Table 6: Evaluation of various models across multiple categories on the T2I-CompBench benchmark.

| Model | Shape | Color | Texture | Spatial Relation | Complex | Average |
|---|---|---|---|---|---|---|
| Best-of-N (N=1) | 64 | 62 | 66 | 70 | 62 | 64.8 |
| Best-of-N (N=2) | 71 | 74 | 75 | 76 | 69 | 73.0 |
| Best-of-N (N=3) | 72 | 78 | 77 | 77 | 70 | 74.8 |
| Ours + FR | 72 | 80 | 81 | 81 | 72 | **77.2** |

a fair and comprehensive evaluation, we report results for $N = 1, 2, 3$ refinement steps in Table 6. It is worth mentioning that parallel TTS methods like repeated sampling can also be combined with our approach.

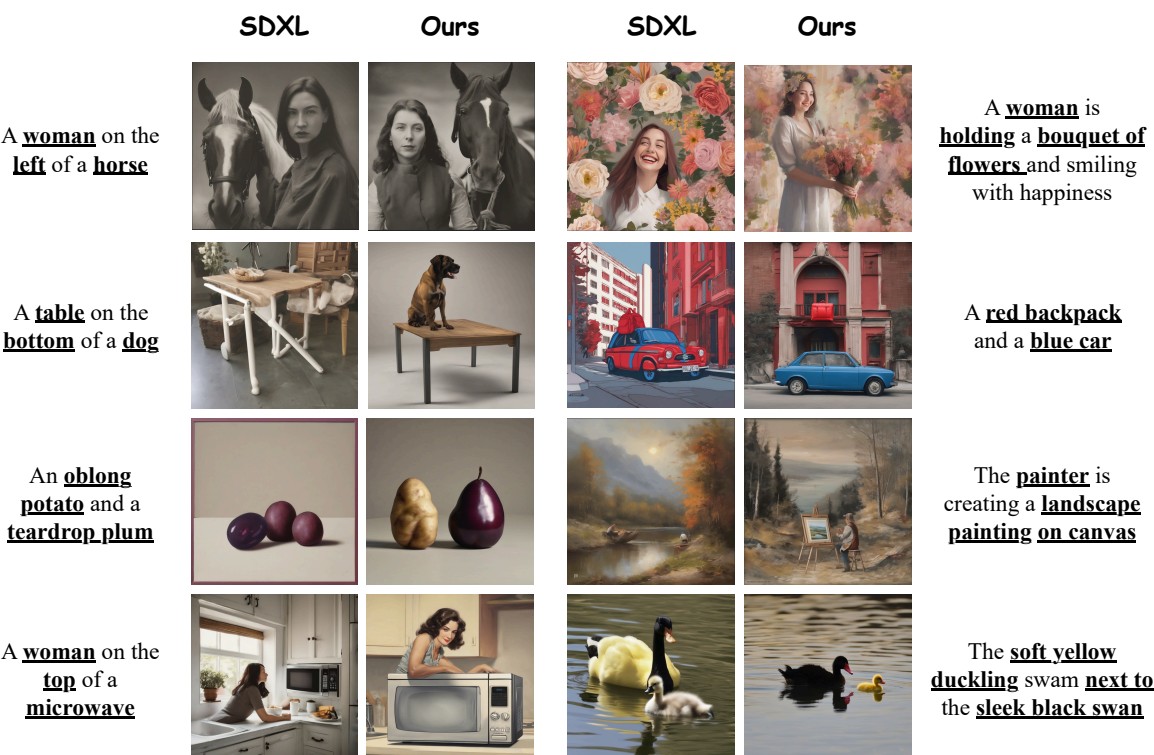

Figure 6: Qualitative comparison of images generated by SDXL against SDXL enhanced with our proposed losses.

## 6.3 Qualitative Results

Figure 5 presents a qualitative comparison of our proposed method against existing models using similar prompts, while Figure 6 compares our approach with the state-of-the-art Stable Diffusion model, SD-XL (Podell et al., 2023). In Figure 5, our method demonstrates superior attribute binding, particularly evident in the shapes depicted in "An oval sink and a rectangular mirror.". Also, regarding spatial relation, all baseline models fail to correctly interpret the concept of *right*, as seen in "A boy on the right of a chicken.". In Figure 6, our method, based on SD v1.4, outperforms SD-XL (Podell et al., 2023) across all categories, including entity missing, attribute binding, and spatial relation. Additional qualitative results are available in Supp. C.5

**Attention maps function as object detection.** In Fig 7, we demonstrate that our last step cross attention maps can be sufficiently rich to serve as object detectors. Comparing our method with Evaluate&Refine (Singh & Zheng, 2023), another soft multi-iteration approach, we show that our method is both faithful and

location-aware, effectively arranging scenes in a coherent manner when prompts contain multiple entities. Further details and visualization can be found in Supp. C.6.

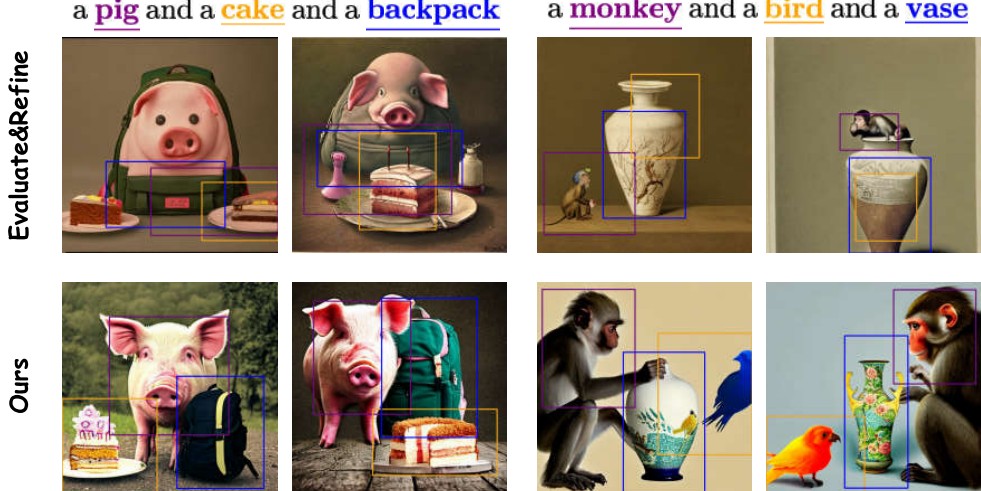

Figure 7: Our Method generates more accurate and distinct attention maps for each entity when the prompt consists of multiple entities. Evaluate&Refine faces the issue of entity mixing (mixing the pig with the backpack) and entity missing (missing the bird), when the number of entities increases, whereas our model is able to separate the entity distributions and create a final coherent image.

## 7 Conclusion

Our work improves text-to-image generative models by addressing compositional challenges through targeted objectives—including entity missing, entity mixing, attribute binding, and spatial relation losses—that enable precise, fine-grained noise adjustments. These interpretable objectives are designed to complement each other and work collaboratively to structure the image scene based on the entities and their relationships mentioned in the prompt. Additionally, our fine-grained initial noise refinement framework employs a feedback-driven system with a verifier that provides fine-grained feedback to separate entities into proper and faulty ones, enabling precise adjustments to the initial noise by targeting only the faulty attention maps while preserving the correct ones. This process iteratively adjusts the initial noise and effectively addresses generation issues with multiple entities.

## 8 Limitations

Our proposed training-free method offers a novel and efficient solution for addressing compositional generation failure modes in T2I models. Nevertheless, it has several limitations that we elaborate on in this section.

**Struggle with long and complex prompts.** As shown in (Zhang et al., 2024a; Liu et al., 2024), diffusion models that rely on CLIP-based text encoders often struggle with long captions. This is because the CLIP text encoder has limited ability to understand and represent the compositional structure of complex or lengthy prompts, often leading to "catastrophic neglect" of secondary details. When the prompt contains several entities (e.g., 4 or more objects with separate attributes), the cross-attention maps associated with later entities become weaker. Earlier entities (and global scene descriptors) dominate the representation, while later entities exhibit lower peak activations and highly overlapping attention with others. Since our method builds on these maps and amplifies the existing signal rather than inventing a new structure, this overlap directly leads to leakage between entities (e.g., attributes from entity A bleeding into entity B), and our optimization cannot reliably disentangle them.

**Non-Optimal Component Design.** Our method introduces an innovative approach by designing a cost function that integrates objectives addressing multiple types of constraints simultaneously. However, our primary goal was not to identify the optimal loss function for each individual constraint (e.g., spatial relationships) or to meticulously tune their relative weights within the overall objective. Instead, we focused on demonstrating the benefits of combining these diverse constraints into a single unified cost function to improve the overall image generation process. As a result, our method may struggle with handling sophisticated instances of compositionality within each component, such as complex spatial relationships (e.g., nested spatial dependencies). This limitation, however, can be addressed in future work by replacing each component with more advanced alternatives.

**Scalability Limitation.** Our method is designed to perform at most two correction iterations. However, when a prompt contains a large number of entities (e.g., four or more), two iterations may not be sufficient to fully address all errors. In such cases, more than two iterations may be necessary to fully correct the output.

**Social Impact.** Our method enhances the control and accuracy of text-to-image generation, which carries significant ethical implications. While advancing creative applications, this work inherits the risks of large generative models, including the potential for misuse and the amplification of societal biases from uncurated training data. We urge that any deployment be coupled with governance and provenance techniques (such as watermarking) to mitigate these issues and ensure responsible application.

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

## A Appendix

## B Verifier Adaptation

We modify the DA-Score (Singh & Zheng, 2023) to provide fine-grained feedback regarding the types of problems present in the image. To do so, we categorize questions by the type of issue associated with each entity. At first, similar to the DA-Score, we extract a diverse set of questions covering all entities. Then, we remove extra details from questions with one entity to create coarse-grained questions to address *entity missing*. Simultaneously, we categorize each question into one of the *attribute binding*, or *spatial relation* problems using LLMs (Achiam et al., 2023). After categorization, we assign a score to each question using a VQA model like DA-Score. For each entity, we aggregate question scores in every category to obtain the final fine-grained feedback. We then use the feedback provided by our evaluator for these questions to assign three scores to each entity, assessing their generation quality in each category.

## C Experiments

### C.1 Benchmarks

We evaluate our work using two benchmarks: T2I-CompBench and HRS.

**T2I-CompBench Benchmark.** T2I-CompBench focuses on compositional text-to-image generation and includes 6,000 text prompts divided into three categories: attribute binding (color, shape, and texture), object relationships (spatial and non-spatial), and complex compositions. As detailed in Section 3.2, the test data was used to assess model performance across various compositional challenges. For a robust evaluation of the spatial relationship category, 150 spatial relationship prompts from the training set were randomly selected, allowing for a diverse assessment using the VISOR Score, as highlighted in Section 3.2.

**HRS-Bench Benchmark.** This benchmark (Bakr et al., 2023) evaluates text-to-image models across 13 skills, grouped into five key areas: accuracy, robustness, generalization, fairness, and bias. Spanning 50 scenarios, including fashion, animals, transportation, food, and clothing, it offers a comprehensive evaluation framework. To demonstrate our model's ability to generate multi-entity images, we utilized the 3-entity prompts from its color category.

### C.2 Metrics

#### C.2.1 Alignment Metrics

We employed various metrics to evaluate our method and compare it with other methods. The most commonly used metric for assessing Text-to-Image models is the CLIP-Score (Hessel et al., 2021), which measures the similarity between the image and text embeddings generated by the CLIP model. The TIFA-Score (Hu et al., 2023) utilizes a language model to generate multiple question-answer pairs from the input prompt, which are then filtered using a QA model. A VQA model answers these visual questions based on the generated image, and the correctness of the answers is evaluated. Additionally, the DA-Score (Singh & Zheng, 2023) breaks the prompt into a set of disjoint assertions and evaluates their alignment with the generated image directly using a VQA model. VQA models often struggle to accurately interpret spatial relationships. We employ position-based metrics as a more dependable approach for evaluating generative models with respect to spatial relationships. We utilize the VISOR Score (Gokhale et al., 2022), designed to assess the spatial relationships of entities. This score is derived by extracting entities using an object detector and calculating the positions of object bounding box centroids relative to one another. The other automated metric we used was the reward model introduced in (Xu et al., 2024). This reward model is trained on human preference comparisons to predict the alignment of generated images with textual prompts, providing an evaluation metric for the quality of text-to-image generation based on human preferences.

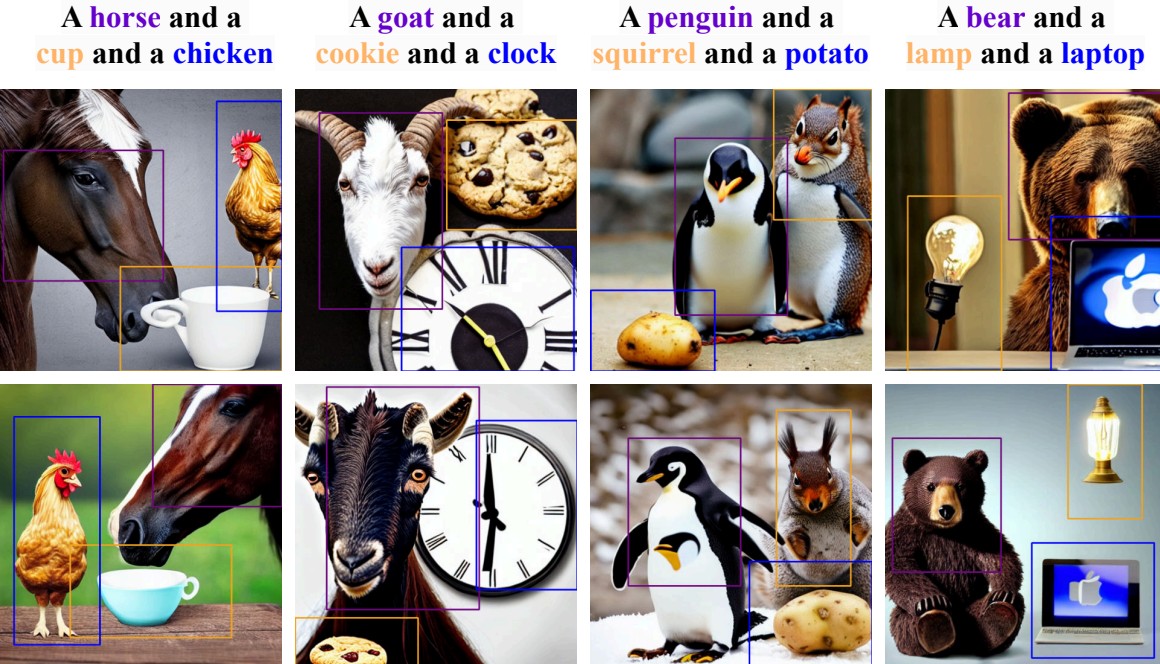

Figure 8: **Our Method generates accurate and distinct attention maps for each entity when the prompt consists of multiple entities.** Entity bounding boxes are extracted from the last attention maps.

### C.2.2 Human Evaluation

We carried out a user study involving 15 volunteers to assess how accurately the generated images corresponded to the text prompts. Each participant chose the image they felt best matched each prompt. Using 25 randomly selected prompts from the T2I-CompBench dataset, we generated images with two different random seeds for each prompt. The alignment score was calculated by counting how often participants preferred images from each model and averaging these preferences across all prompts. We provided volunteers with a set of guidelines to follow. First, they were instructed to select images where no entities were missing. As the second priority, they were asked to choose images with accurate attribute binding and spatial relationships. Lastly, they considered how well the image aligned with the prompt and whether its overall quality was better than the alternatives.

### C.3 More ablation Study

### C.3.1 Fine-grained Refinement Method Ablation Study

We conducted an experiment to assess the impact of our multi-object approach and our multi-objective approach, including *attribute binding* and *spatial relation* objectives. Fig. 9 demonstrates that the omission of each concept leads to a substantial decline in accuracy, showcasing their contribution to the entire framework. We utilized the HRS 3-entity prompts, indicating the effectiveness of our method.

### C.3.2 Diversity & Reality

To evaluate the realism and diversity of generated images and compare models, we used the *Density* and *Coverage* metrics proposed by (Naeem et al., 2020). *Density* measures fidelity by assessing how many real data points lie within the neighborhood of each generated sample, reflecting how well the generated data aligns with the real distribution. *Coverage* quantifies diversity by determining the proportion of real data

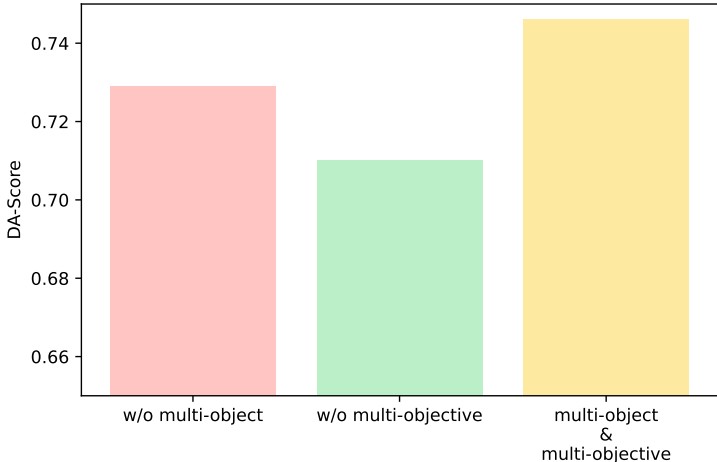

Figure 9: Ablation study on fine-grained refinement method. We evaluated on only correcting a single entity (w/o multi-object) and addressing only the entity missing issue (w/o multi-objective). The results are taken on the HRS dataset.

Table 7: Comparison between three different losses—two from previous works (Kullback–Leibler divergence and Jensen–Shannon divergence) and our proposed IoU-based loss to address the *attribute binding* loss.

| Loss Term | | | Compositional Categories | | | | |
|---|---|---|---|---|---|---|---|
| KL | JSD | IoU(Ours) | Shape | Color | Texture | Complex | **Average** |
| ✓ | | | **0.67** | 0.76 | 0.78 | 0.69 | 0.73 |
| | ✓ | | **0.67** | 0.78 | 0.77 | 0.68 | 0.73 |
| | | ✓ | 0.66 | **0.79** | **0.81** | **0.71** | **0.74** |

points that have at least one generated sample in their neighborhood, indicating how effectively the model captures the variability of real data.

## C.4   Additional Quantitative Results

### C.4.1   Attribute Binding Loss Comparison

We evaluated three different losses for attribute binding. As shown in Table 7, this analysis demonstrates the effectiveness of our IoU-based loss for addressing the attribute binding problem, outperforming other losses employed in previous studies (Li et al., 2024b).

### C.4.2   Additional Metrics

As presented in Table 8, our method surpasses all other models on the Image-Reward metric for the HRS benchmark and most categories of the T2I-CompBench benchmark while achieving comparable performance in the *Shape* and *Non-spatial* categories.

Similarly, as demonstrated in Table 9, our method outperforms all other models on the CLIP-Score metric for the HRS benchmark and most categories of the T2I-CompBench benchmark, while achieving comparable performance in the "Complex" and "Texture" categories. On the other hand, this metric struggles to underscore the differences between various methods, as the score variations among different models are smaller compared to other metrics. This limitation arises from the inherent weaknesses of the CLIP (Radford et al., 2021) model in compositional understanding, primarily due to its training methodology and the scoring mechanism, which overlooks several aspects of compositionally.

| A **bird** on **top** of a **baloon** | A **suitcase** on the **bottom** of a **turtle** | A **boy** on the **right** of a **chicken** | A **woman** on the **left** of a **cup** |

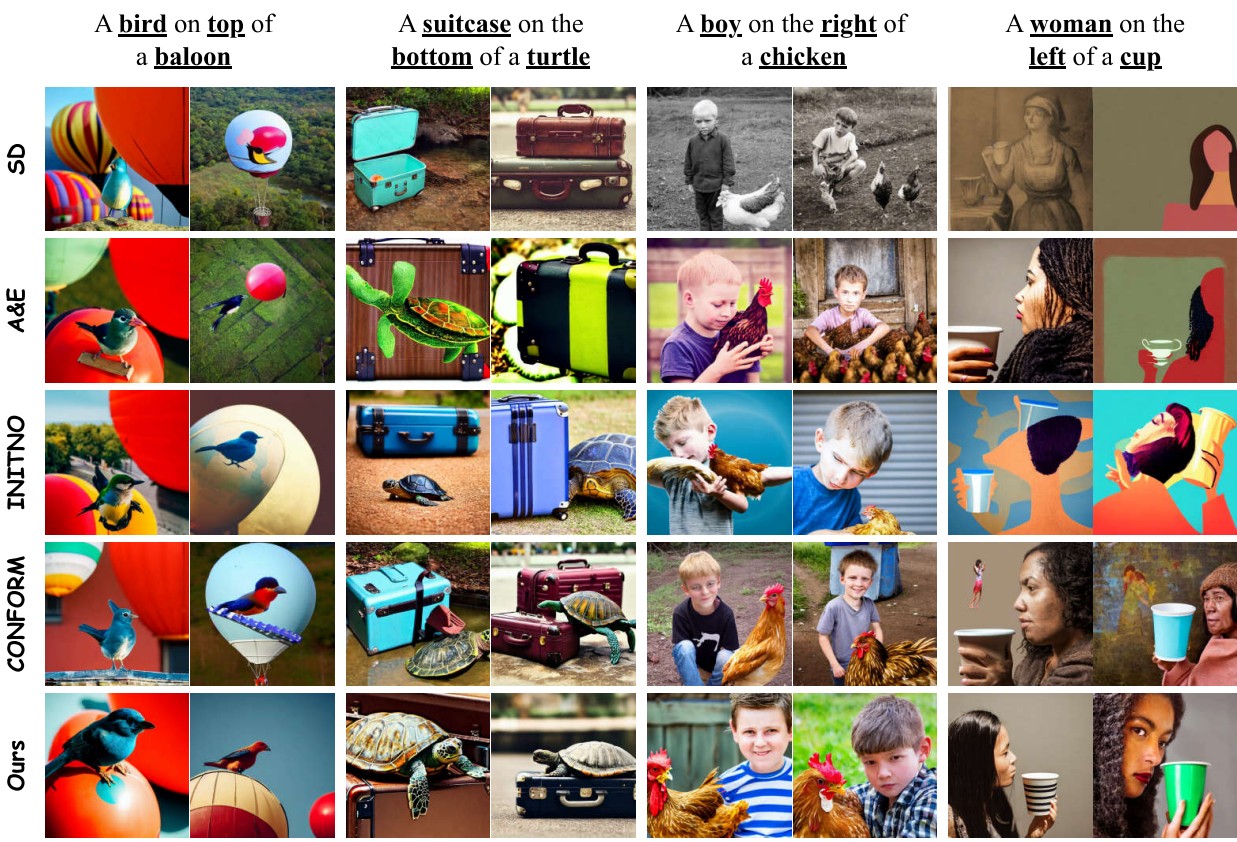

Figure 10: Qualitative comparison of our method with state-of-the-art models on spatial prompts. Our model demonstrates a consistent and superior understanding of spatial relation complexities.

Table 8: **Performance Comparison**. Evaluation of various models on the Image-Reward metric. We assessed across different categories of the T2I-CompBench benchmark along with the HRS benchmark.

| Model | Color | Complex | Relation | | Shape | Texture | HRS |
|---|---|---|---|---|---|---|---|
| | | | **Spatial** | **Non-spatial** | | | |
| Stable v1.4 | -0.16 | -0.24 | 0.15 | 0.49 | -0.43 | -0.53 | -1.35 |
| Stable v2 | -0.02 | 0.31 | 0.22 | **0.59** | **-0.01** | -0.26 | -1.04 |
| A&E | 0.41 | 0.13 | 0.73 | 0.18 | -0.42 | -0.18 | -0.06 |
| D&B | 0.02 | 0.13 | 0.51 | 0.49 | -0.25 | 0.00 | -0.64 |
| Evaluation&Refine | -0.07 | -0.17 | 0.21 | 0.57 | -0.61 | -0.11 | -1.49 |
| CONFORM | 0.37 | -0.03 | 0.89 | 0.18 | -0.28 | 0.55 | -0.27 |
| INITNO | 0.60 | -0.08 | 0.73 | 0.39 | -0.02 | 0.70 | 0.18 |
| Ours | 0.76 | 0.42 | 0.80 | 0.26 | -0.16 | 0.62 | 0.26 |
| Ours + FR | **0.96** | **0.44** | **0.93** | 0.47 | -0.02 | **0.71** | **0.54** |

Nevertheless, we employ various objective functions and utilize detailed textual information to arrange the scene effectively. As shown in Table 10, our generated images surpass other models in diversity, as measured by the *Coverage* metric. Additionally, we achieve results comparable to the best-performing models in the *Density* metric, which determines the realism, and the *Aesthetic Score*, reflecting overall quality.

Table 9: **Performance Comparison**. Evaluation of various models on the CLIP-Score metric across different categories using the T2I-CompBench benchmark and HRS benchmark.

| Model | Color | Complex | Relation | | Shape | Texture | HRS |
|---|---|---|---|---|---|---|---|
| | | | Spatial | Non-spatial | | | |
| Stable v1.4 | 0.325 | 0.301 | 0.318 | 0.303 | 0.308 | 0.312 | 0.32 |
| Stable v2 | 0.322 | **0.306** | 0.319 | 0.304 | 0.313 | 0.303 | 0.33 |
| A&E | 0.331 | 0.300 | 0.328 | 0.304 | 0.304 | 0.302 | 0.34 |
| D&B | 0.325 | 0.301 | 0.316 | 0.305 | 0.310 | 0.310 | 0.33 |
| Evaluation&Refine | 0.321 | 0.301 | 0.318 | 0.307 | 0.296 | 0.310 | 0.30 |
| CONFORM | 0.320 | 0.299 | 0.330 | 0.304 | 0.305 | 0.314 | 0.34 |
| INITNO | 0.332 | 0.296 | 0.324 | 0.301 | 0.311 | **0.324** | 0.34 |
| Ours | 0.333 | 0.303 | 0.324 | 0.305 | 0.309 | 0.322 | 0.34 |
| Ours + FR | **0.335** | 0.302 | **0.326** | **0.308** | **0.314** | 0.322 | **0.35** |

Table 10: Evaluation of image realism and diversity.

| Model | Coverage ↑ | Density ↑ | Aesthetic Score ↑ |
|---|---|---|---|
| Stable v1.4 | 0.76 | **0.67** | 5.882 |
| Stable v2 | 0.73 | 0.46 | **5.924** |
| A&E | 0.77 | 0.56 | 5.684 |
| D&B | 0.76 | 0.63 | 5.764 |
| Evaluation&Refine | 0.72 | 0.44 | 5.594 |
| CONFORM | 0.69 | 0.43 | 5.533 |
| INITNO | 0.76 | 0.61 | 5.668 |
| Ours | 0.76 | 0.60 | 5.703 |
| Ours + FR | **0.79** | 0.65 | 5.712 |

### C.4.3 Impact of threshold

We provide an ablation study on the impact of the threshold used for including or omitting losses in the final loss term. As shown in Figure 11, this hyperparameter has minimal impact on the average impact across all the categories. This means the model is not sensitive to this value, and this is due to the overconfidence of the question-answering model, which assigns high probability to the dominant class.

### C.5 Additional Qualitative Results

Figure 10 presents a side-by-side comparison of our method with state-of-the-art models, focusing exclusively on spatial relations, while Figure 15 compares our approach with other baselines across all categories. These figures highlight common failure cases in existing models that our method effectively addresses, such as entity missing in "a blue giraffe and a brown vase" and attribute binding errors like "a yellow banana and a red monkey." Additionally, our method successfully handles complex spatial relationships where other models struggle, as demonstrated in "a brown teddy bear holding a blue cup sits on the sofa."

**Comparison of our models with fine-tuned models** We compare our models with GORS (Huang et al., 2023a), a method that fine-tunes Stable Diffusion v2 using generated images closely aligned with compositional prompts. In this approach, the fine-tuning loss is weighted by a reward based on alignment scores. As demonstrated in Fig. 12, our models outperform GORS on the T2I-CompBench (Huang et al., 2023a).

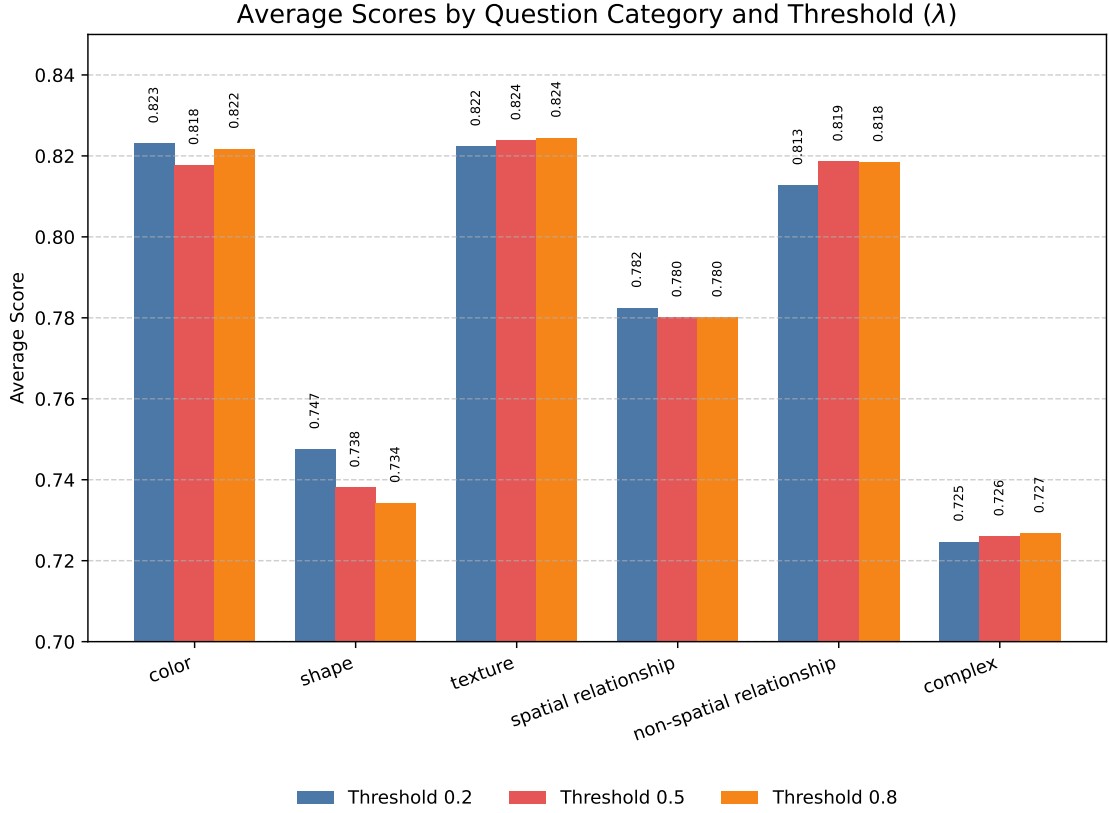

Figure 11: Ablation study on the impact of the loss inclusion threshold ($\lambda$) on the results across all categories.

**Visualization of the attention maps.** Given the prompt *a black cat is on the left of a green frog*, we visualize the final cross-attention map for each subject and attribute token after the denoising process in Fig. 13. The left side shows the output of Stable Diffusion v1.4, while the right side presents the results of our fine-grained initial noise refinement. Our method achieves more precise attention allocation, ensuring semantically consistent outputs. Unlike Stable Diffusion, where attention is dispersed across the entire scene, our approach focuses attention on each entity and its corresponding attribute.

**Failure Case Study.** We illustrate the shortcomings of our methods with three prompts under different categories in Fig. 14. In the first example, the high number of entities results in a race between losses of each entity. In result, some of them will have too little of a contribution to the final loss and will be ignored and the object is not fixed in the result. This is a kind of failure mode where the verifier performed well but our losses could not replicate the complexity of the scene. In the second example, we see an example of nested relations (see section 8). The verifier can not pinpoint the exact relation between entities which results in the missing of object. In the last example as we expect, VQA models drop performance when faced with long and complex prompts. In these scenarios, the verifier fails to extract the correct sense of image structure in the prompt. Similar to the second example, the verifier error propagates into the correction stage and results in images with mistakes.

### C.6 Attention maps as object detection

We extracted object boxes from attention maps to qualitatively assess the generated images. The details for bounding box extraction are provided in section D. Fig. 8 illustrated some more visualizations of entity boxes from our generated images. Extracted boxes show great alignment with the location where the image is actually generated, showcasing the proper appearance of multiple entities.

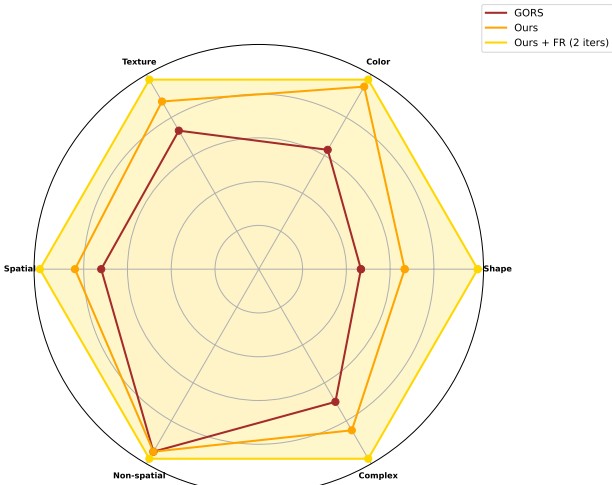

Figure 12: **Performance Comparison.** Comparison of our models with GORS, as figure shows our models outperform GORS on the T2I-CompBench dataset across all categories, demonstrating superior alignment between generated images and compositional prompts.

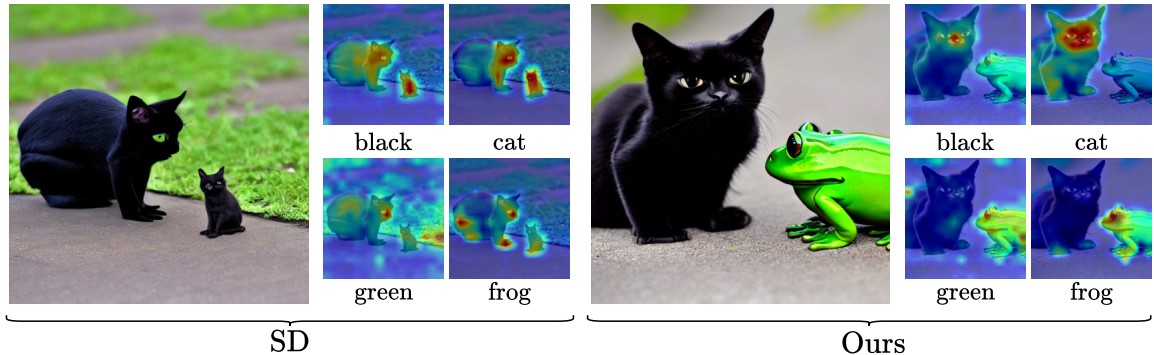

Figure 13: **Visualization of the final cross-attention maps.** Compare the final cross-attention maps of Stable Diffusion v1.4 and our fine-grained initial noise refinement. The prompt is: "A black cat is on the left of a green frog."

Table 11: **Performance Comparison**. Evaluation of the impact of our objective functions during inference of XL Stable Diffusion model on four different metrics.

| Categories | Image-Reward | | DA-Score | | TIFA-Score | | CLIP-Score | |
|---|---|---|---|---|---|---|---|---|
| | SDXL | Ours | SDXL | Ours | SDXL | Ours | SDXL | Ours |
| Spatial | 0.816 | 0.973 | 76 | 77 | 82 | 85 | 0.330 | 0.333 |
| Color | 0.612 | 0.682 | 77 | 75 | 86 | 83 | 0.336 | 0.340 |
| Complex | 0.269 | 0.249 | 69 | 70 | 77 | 78 | 0.305 | 0.304 |
| Non-Spatial | 0.652 | 0.754 | 80 | 81 | 87 | 89 | 0.308 | 0.308 |
| Shape | 0.164 | 0.060 | 68 | 67 | 69 | 71 | 0.309 | 0.313 |
| Texture | 0.235 | 0.232 | 75 | 76 | 82 | 84 | 0.330 | 0.334 |

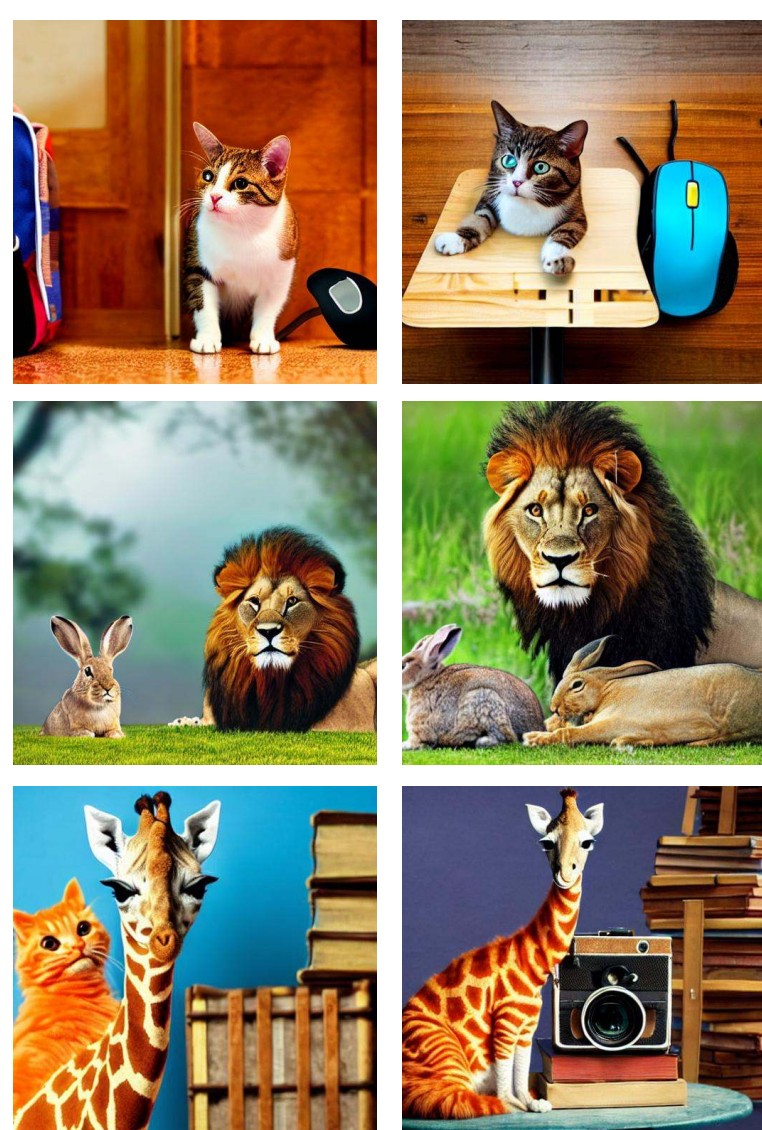

a cat and a table and a
mouse and a backpack.

a lion is resting,
while a rabbit is
sitting near the lion,
and a bird is
flying above the rabbit

a tall giraffe is standing
beside a wooden crate,
a fluffy orange cat is sitting
near a stack of old books,
and a small blue bird is perched
on a vintage camera
placed on a round table

Figure 14: **Failure Case Study.** Illustration of FR failure cases. each prompt is generated across two different seeds. By pattern either the validator makes mistakes or the optimization problem even with different losses is too hard.

## D Implementation Details

We implement our method using the official Stable Diffusion v1.4 text-to-image model. Following the Attend-and-Excite approach, we apply a fixed guidance level of 7.5 and set the scale factor $\alpha_t$ with a linear schedule starting at 20 and linearly decreasing to a minimum of 10. The threshold for identifying problematic entities ($\lambda$) is set at 0.5. We set the denoising step count, $T$, to 50, and conduct image generation on an RTX 3090 GPU. For loss calculation, we first normalize the outputs of the cross-attention layers to a range between 0 and 1. In our optimization setup, we assign equal weights to different loss terms without any hyperparameter tuning. Lastly, our quantitative results are calculated across multiple random seeds.

**Details for Extracting Bounding Boxes.** To extract object boxes, we first upscale the attention maps to match the image resolution. We then apply a Gaussian blur to smooth it. After normalizing the attention

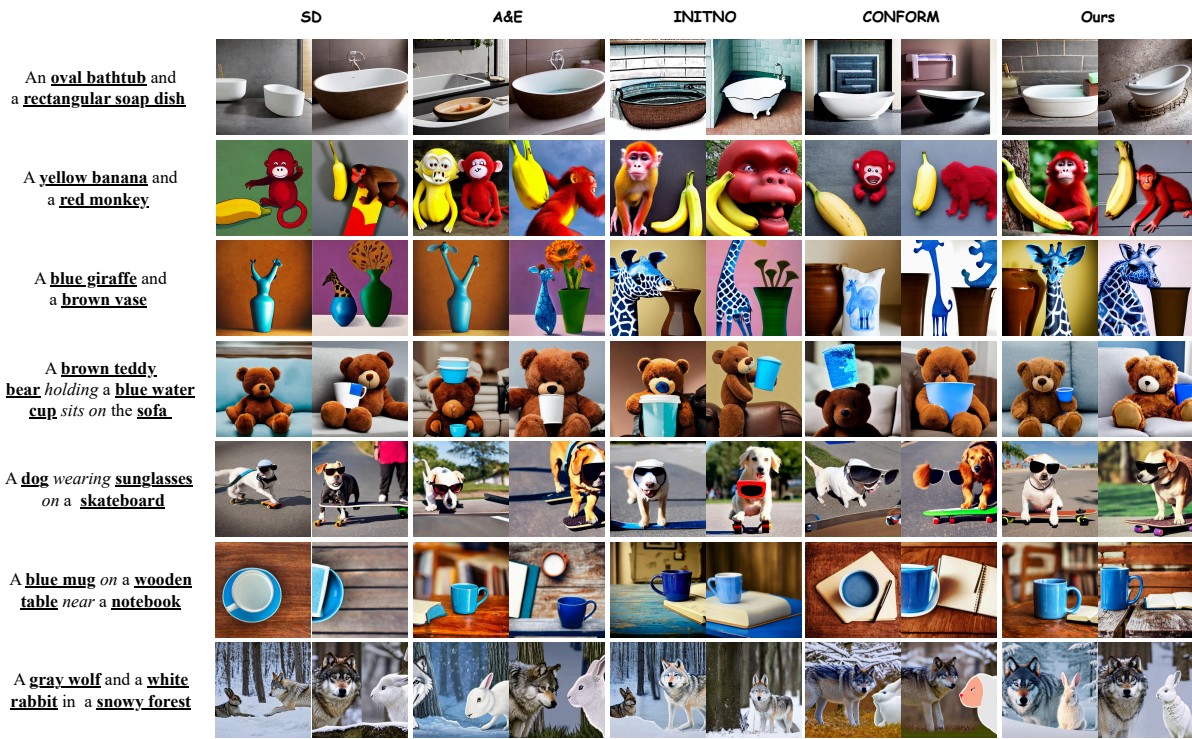

Figure 15: We generated two images for each of the seven example text prompts, using different random seeds for a qualitative evaluation of our approach compared to other models.

maps using the max-min method, we then select the top 10% pixels with the highest value, and the largest contour is selected as the entity bounding box.

**Prompt Decomposition.** Our objective functions and our framework require entity, attribute, and relation sets for supervision. To extract these sets, we query GPT-4o with 3 decomposition samples to enforce correct task achievement. 16 and 17 provide an overview of the different prompt decomposition outputs for our approach.

Given a caption:
Specify the spatial location among objects in the caption as (object, spatial relation, object). If there is not any spatial relation in the caption, just say "No spatial"

Example:
Caption: "the striped rug was on top of the tiled floor"
1. (rug, on top of, floor)

Example:
Caption: "a couple is enjoying a picnic in the park"
1. (couple, in, park)

Example:
Caption: "a blue scooter is parked near a curb in front of a green vintage car"
1. (scooter, near, curb)
2. (scooter, in front of, car)

Example:
Caption: "the airplane is flying above the clouds."
1. (airplane, above, clouds)

Example:
Caption: "a bird on the left of a clock"
1. (bird, on the left of, clock)

Example:
Caption: "the black phone was resting on the silver charger"
1. (phone, on, charger)

Example:
Caption: "the bear is near a lake"
1. (bear, near, lake)

Example:
Caption: "a mouse on side of a bag"
1. (mouse, on side of, bag)

Example:
Caption: "a small white kitchen with brown wood floor"
No spatial

Answer as concisely as possible.

Figure 16: The few-shot prompt used for extracting spatial relations from the text.

Given a caption:
Specify the adjective and its object in the caption as (adjective, object) If there is not any adjective in the caption, just say "No adjective"

Example:
Caption: "a brown dog fetching a frisbee in a green park."
1. (brown, dog)
2. (green, park)

Example:
Caption: "a long necklace and a short earring"
1. (long, necklace)
2. (short, earring)

Example:
Caption: "a soft pillow is on top of the rocking chair"
1. (soft, pillow)
2. (rocking, chair)

Example:
Caption: "a metallic bracelet and a leather hat"
1. (metallic, bracelet)
2. (leather, hat)

Example:
Caption: "a person is wearing a hat and sunglasses while fishing"
No adjective

Example:
Caption: "a desk on the right of a horse"
No adjective

Example:
Caption: "the shiny silver watch lay next to the smooth black leather wallet"
1. (shiny, watch)
2. (silver, watch)
3. (smooth, wallet)
4. (black, wallet)
5. (leather, wallet)
Answer as concisely as possible.

Figure 17: The few-shot prompt used for extracting attribute-entity pairs from the text.

