# OpenReview forum: "Fine-Grained Alignment and Noise Refinement for Compo- sitional Text-to-Image Generation"
_TMLR — Rejected by TMLR_

### Review · Reviewer_fpMx · 2025-08-19

**Summary Of Contributions:**

This paper introduces a novel, training-free method to enhance compositional text-to-image generation in diffusion models. The core contribution is a unified loss function, termed EAR loss, which simultaneously addresses four key challenges: entity missing, entity mixing, attribute binding, and incorrect spatial relationships. This is achieved by manipulating cross-attention maps during the inference process to align the generated image more closely with the textual prompt. A second major contribution is a feedback-driven, fine-grained initial noise refinement framework. This two-stage process first generates an image and uses a "verifier" model to detect misalignments. It then uses this feedback to selectively refine only the faulty aspects of the initial noise, preserving correctly generated elements, before proceeding to a second generation phase.



## Strengths

- The work tackles multiple critical failure modes of text-to-image models within a single, unified framework, which is a significant step beyond prior methods that often focus on only one or two of these issues.

- The method works at inference time and does not require costly retraining or fine-tuning of the diffusion model, making it highly practical and accessible.

- The idea of selectively refining parts of the initial noise based on feedback is novel and more efficient than methods that regenerate the entire noise vector.

## Weaknesses

- The framework's effectiveness relies on the performance of an external VQA model (as a verifier) and an LLM for prompt decomposition. The potential for errors or biases from these components to propagate is a limitation.

- While training-free, the full two-stage method involves multiple components (generation, verification, selective loss application, re-generation) that add complexity to the inference pipeline.

**Audience:**

Yes

**Audience Explanation:**

This paper will be of interest for several reasons:

- It addresses a fundamental and unsolved problem in a popular and rapidly advancing area of research.

- The proposed training-free solution is practical and can be readily applied to existing pre-trained models, making it appealing to a broad audience, including those without the resources for large-scale model training.

- The detailed analysis and strong empirical results provide valuable insights and a new state-of-the-art benchmark for compositional text-to-image generation.

**Claims And Evidence:**

Yes

**Claims Explanation:**

The claims are well-supported by a comprehensive set of experiments and evaluations.

- The authors conduct extensive quantitative comparisons against nine baselines (including base models like Stable Diffusion v1.4/v2/XL and state-of-the-art training-free methods like CONFORM and INITNO) on two established benchmarks (T2I-CompBench and HRS). The results, measured by multiple automated metrics (DA-Score, TIFA-Score, VISOR Score), consistently show that the proposed method, particularly with fine-grained refinement (FR), outperforms existing approaches across nearly all compositional categories. For instance, the claimed 25% improvement in spatial relationships is clearly demonstrated by the VISOR Score results in Figure 4(a).


- The paper includes targeted ablation studies (Table 6) that effectively isolate and validate the contribution of each component of the EAR loss. This demonstrates that the different loss terms are complementary and necessary for the model's overall performance.

- A user study involving 15 participants further strengthens the claims, showing a clear human preference (a 24% improvement) for the images generated by the proposed method in terms of text-image alignment. This is crucial evidence, as automated metrics can sometimes fail to capture human perception of quality and accuracy.

**Requested Changes:**

- In Section 5.2, the correction loss (Equation 10) uses an indicator function. The implementation details state λ is set to 0.5, but there is no justification for this choice. Please add a brief discussion on how this threshold was determined. Is the model's performance sensitive to this value? Providing this information is critical for the method's reproducibility.

- The appendix mentions that the DA-Score verifier was modified to provide fine-grained feedback by categorizing questions using an LLM. This is a key part of the feedback mechanism. Please provide more detail on this process. For example, what prompt was used for the LLM-based categorization, and how are the final entity scores aggregated from the VQA outputs? This would improve clarity and help others build upon your work.

- The limitations section rightly points out that the method inherits the struggles of CLIP-based encoders with long, complex prompts. It would strengthen the paper to briefly elaborate on why this is the case within your framework. For instance, do the attention maps become less distinct for a higher number of entities, or does the LLM-based decomposition start to fail? A more detailed diagnostic could provide valuable insights for future work.

- The paper shows many successful examples. However, are there cases where the fine-grained refinement fails or even degrades the image quality? Including a brief discussion or a visual example of a failure case (e.g., the verifier provides incorrect feedback, leading to a flawed correction) would provide a more balanced view of the method's capabilities.

---

> ### Author Response · Authors · 2025-12-08
> **Response**
>
> Thank you for your thoughtful comments and suggestions.
>
> ### Correction Loss Threshold Justification:
> We set the faulty threshold to 0.5 because the DA-Score for each <image, question> pair returns a value in [0, 1], representing the probability of a ‘Yes’ answer. Values above 0.5 indicate correct, while values below 0.5 indicate faulty ones. To further address the reviewer’s concern, we have included a sensitivity analysis of the faulty threshold in Appendix C.4.3 of the revised paper.
>
>
> ### Fine-Grained Feedback Mechanism Details:
> The prompts used for LLM-based categorization are provided in Appendix D under ‘Prompt Decomposition.’ Please refer to Figures 14 and 15 for the specific prompts used to extract attributes and relationships from each input prompt.
>
>
> Regarding entity score aggregation, for each given prompt we generate a single entity, attribute, and spatial question, each of which is scored using the modified DA-Score (see Appendix B for scoring details). These scores are then used to activate their corresponding losses, which are finally aggregated according to Eq. 10.
>
>
> ### Long/Complex Prompts Limitation:
> Thank you for your question, which helped us clarify the limitations of our method with respect to long and complex prompts in the revised paper. As discussed in limitations, Stable diffusion models rely on the pre-trained CLIP text encoder, which introduces inherent bottlenecks for long or complex prompts. We attribute this performance degradation to two key factors identified in recent literature:
>
>
> - **Effective Context Length:** As demonstrated in [1], although CLIP has a theoretical limit of 77 tokens, its effective information retention is often fewer than 20 tokens. Consequently, constraints for entities appearing later in a long prompt are often encoded with negligible signal strength, making their attention maps too faint for our optimization to effectively segregate.
>
>
> - **Alignment Degradation:** As detailed in [2], standard CLIP encoders struggle to maintain semantic alignment across lengthy descriptions, often leading to "catastrophic neglect" of secondary details. When the prompt contains several entities (e.g., 4 or more objects with separate attributes), the cross-attention maps associated with later entities become weaker. Earlier entities (and global scene descriptors) dominate the representation, while later entities exhibit lower peak activations and highly overlapping attention with others. Since our method builds on these maps and amplifies the existing signal rather than inventing a new structure, this overlap directly leads to leakage between entities (e.g., attributes from entity A bleeding into entity B), and our optimization cannot reliably disentangle them.
>
> [1] Zhang, B., et al. “Long-CLIP: Unlocking the Long-Text Capability of CLIP,” in ECCV, 2024.
>
> [2] Liu, L., et al. “Improving Long-Text Alignment for Text-to-Image Diffusion Models,” in ICLR, 2025.
>
>
> ### Failure Case Discussion:
> We appreciate the suggestion to include failure cases. We have added general failure cases of our method in Appendix C.5 and Figure 14.
> As shown in the first case, when a high number of entities contribute to the loss, they "race" against each other, causing some entities' losses to be too small and effectively ignored. So the loss function can not adequately capture the scene's complexity. The issue is with the loss calculation, not the verifier's performance. In the second and third example, as expected, VQA models exhibit reduced performance when faced with long and complex prompts. Specifically the second example focuses on the nested relationship problem which was highlighted in Section 8 (Limitations). In these scenarios, the verifier fails to correctly capture the intended image structure from the prompt. This verifier error propagates to the correction stage, resulting in images that contain mistakes.
>
>
> ### Weaknesses:
> Thank you for your valuable feedback.
> We acknowledge that our method is subject to limitations inherent in the performance of the VQA model. However, we anticipate that as VQA models and LLMs continue to improve, their contribution to errors will diminish over time. To address this, we have included an analysis of VQA-related errors in the newly added Failure Case Study subsection (Appendix C.5).
> With respect to the complexity of our method, while we recognize that it introduces additional steps, we can see that it improves inference speed relative to other multi-iteration baselines and achieves comparable speed to optimization-based methods as shown in Table 5. Notably, this increase in complexity does not hinder speed but rather leads to enhanced performance.

---

### Review · Reviewer_77U9 · 2025-09-08

**Summary Of Contributions:**

This paper proposes a set of differentiable penalties for guiding cross attention maps during sampling, with "losses" targeting Entity errors, Attribute binding errors, and Relationship errors (EAR). Latents are updated by one step from the cross attention scores, and 4 terms are combined to produce the signal for the update, similar in spirit to classifier-free guidance.

As this reviewer understands the work; after the initial EAR pass, a verifier flags which entities failed, and how (missing, mixing, attribute, spatial). The initial noise is then optimised with a gated objective using the evidence from the verifier, including a preservation loss using IoU over current and reference attention maps.

Combined, the approach is using the same geometric constraints twice, at two different places in the sampling procedure, with different scopes. Overall, the paper targets existing pipelines with a small improvement that could prove particularly useful in several cases where prompt engineering is not enough to improve the quality of reconstructions alone. The approach is lean and cost effective, while showing some modest empirical improvements.

Limitations are addressed, but seems to be broad and general; see next section and requested changes for details.

**Audience:**

Yes

**Audience Explanation:**

The study focuses on a low cost plug-and-play framework which can improve consistency in T2I approaches which would be of interest to generative approaches and multimodal alignment in general. The approach is general, and can be applied with different T2I models and backbones. As this is a current frontier in research, this reviewer believes the paper is of interest to several readers in TMLR's audience. However, there are some issues related to what extent the work can be applied, as discussed in the previous part on claims, particularly since the approach is framed as a "non-learning" approach. In these works, the modeling choices should be well motivated and clear to appeal to a broader audience.

**Broader Impact Concerns:**

Generative T2I approaches warrants a broader impact concern, since they can be applied to more convincingly generate images that could have ethical concerns. This needs to be included and discussed in the paper.

**Claims And Evidence:**

No

**Claims Explanation:**

The study provides empirical evidence for their practicality of the work and the claims, including several metrics alongside a qualitative human study. The experiments are clearly laid out and well documented. Limitations are laid out, and the authors discuss issues with complex prompts, component design, scalability and cost / compute. Results are modest but consistent.

However, it should be stated that the limitations should be more clearly connected to the results in the paper. Which exact modeling choices are the components facing limitations related to? The limitations come across as very broad in the current manuscript. In some respects, there are several modeling decisions that seemed "pulled from a magical hat" and seems to not be thoroughly motivated. In this sense, the work reads as slightly unsatisfying, since the mechanics are not clearly laid out and offers little insight into what modeling choices impact the results. Ablations are very sparse, and could be more clearly motivated either theoretically or empirically.

In this reviewers opinion, while central claims are corroborated with the empirical evidence, the authors lack evidence or motivation for several of the modeling choices. This is a point where the authors could improve their work for publication.

**Requested Changes:**

1. Calling the EAR terms "losses" is a bit misleading; they are arguably inference-time regularisation terms, and calling them as such could improve legibility. At the very least, making a clear distinction between an inference objective and a training objective would improve the paper, and should be clarified in the article. Granted, this is a smaller issue than some of the others.
2. There are a lot of modeling choices that are not clearly motivated;
    - (a) Min-max scaling of attention maps; not clear why this is a necessity. Could the authors expand on this and argue for why this was selected? Typically attention maps are already implicitly scaled by softmax scores, so this point is a bit confusing and warrants more discussion in the manuscript or elaboration on the authors part.
    - (b) Constraining the attention maps to $16 \times 16$ latent dimensions seems somewhat limiting. The faithfulness of such a generalisation should be handled with slightly more care in this reviewers opinion, however, this is discussed as a limitation by the authors in non-optimal component design. An ablative study with small increments would more convincingly show that this limitation is inherently coupled with scaling, and not a matter of concern in further development of the idea.
    - (c) There are some issues with the preservation term using IoU; why was this selected? Attention IoU seems like a design choice that may be a source of improvement, and while it seems to work, it is not ablated. What other metrics were considered? Should this be considered as one of the limitations?
    - (d) Another choice is on the spatial loss. The logistic sigmoid has a tendency to saturate. When applied to center-of-mass differences, this could imply that large violations gives very small gradients when stronger corrections are required. How does this affect the regularization with EAR?
3. The discussion on inference time should be reported in a table, or be included in the existing tables (1,2) to let readers actually compare the results. While addressed somewhat in its own section, it should be emphasised and be clear to the reader when comparing the results. Particularly, leading with non-FR result seems a bit misleading since the FR approach is touted as the top performing model / approach.
4. in Figure 7 and 8; what exactly are the authors demonstrating? Is this supposed to illustrate a qualitative improvement? Which method is which? What exactly are the authors comparing against to state that "their method is more accurate"? This should either be removed or the authors should rewrite the caption such that it is clear what they mean by this. It seems cherry-picked with no actual explanation.

---

> ### Author Response · Authors · 2025-12-08
> **Response-Part1**
>
> ## EAR Terminology:
> Thank you for this important note on terminology. We agree that the terms are more accurately described as **inference-time regularization objectives**, as they are training-free and do not update model weights. However, we have chosen to use the widely adopted term **loss** for two main reasons:
> - **Consistency with Prior Works**: This terminology is widely accepted and used in the literature on Text-to-Image models [1, 2, 3].
> - **Functional Clarity**: Operationally, these terms are objectives that are minimized via optimization (gradient descent) at each refinement step, and this minimization is consistent with the concept of a loss function as defined in Machine Learning.
>
> Alongside these, we have repeatedly mentioned in the paper that our approach is training-free, ensuring readers do not confuse the loss functions with training objectives. However, if you believe this change is necessary, we will modify the paper accordingly.
>
> [1] H. Chefer et al., “Attend-and-Excite: Attention-Based Semantic Guidance for Text-to-Image Diffusion Models,” in ACM Trans. Graph. 2023.
> [2] A. Agarwal et al., “A-STAR: Test-time Attention Segregation and Retention for Text-to-Image Synthesis,” in ICCV, 2023.
> [3] X. Guo et al., “InitNo: Boosting text-to-image diffusion models via initial noise optimization,” in CVPR, 2024.

---

> ### Author Response · Authors · 2025-12-08
> **Response-Part2**
>
> ## Modeling Choices Motivation/Ablation:
>
> ### (a) Min-max scaling of attention maps:
>
> Thank you for your feedback. We included a citation to the paper we were inspired by to clarify this design choice.
>
> Attention maps are normalized over the token dimension using softmax; however, this does not normalize each token’s 2D attention map. Independent normalization within each map is essential for properly highlighting the most important pixels. Therefore, pixel-wise normalization is required. In addition, the choice of Min-Max scaling is inspired by recent state-of-the-art compositional methods [1], which use min-max normalization on the attention maps at each step. We added citation to [1] in section 4 (Definitions).
>
> [1] Guo, Q., et al.  “Focus on Your Instruction: Fine-grained and Multi-instruction Image Editing by Attention Modulation”, in CVPR, 2024.
>
> ### (b) Constraining attention maps to $16\times16$ latent dimensions:
> There are two reasons for choosing this resolution:
> 1. It corresponds to the deepest cross-attention blocks, where high-level semantic layout and global structure are determined.
> 2. Using this resolution is considered best practice and is adopted by other state-of-the-art models in the field [1, 2, 3, 4].
>
> We had already included a reference to [4], where we introduced the $16 \times 16$ attention maps at the end of the Preliminaries section.
>
> [1] H. Chefer et al., “Attend-and-Excite: Attention-Based Semantic Guidance for Text-to-Image Diffusion Models,” in ACM Trans. Graph. 2023.
>
> [2] A. Agarwal et al., “A-STAR: Test-time Attention Segregation and Retention for Text-to-Image Synthesis,” in ICCV, 2023.
>
> [3] X. Guo et al., “InitNo: Boosting text-to-image diffusion models via initial noise optimization,” in CVPR, 2024.
>
> [4] Hertz, A., et al. “Prompt-to-Prompt Image Editing with Cross Attention Control,” arXiv, 2022.
>
> ### (c) Preservation term using IoU:
> We have made a comparison between IoU and similar metrics in **Appendix C.4.1** and **Table 9** of the submission. As detailed in that section, we quantitatively compared IoU against alternative metrics, specifically **KL Divergence** and **Jensen-Shannon Divergence (JSD)** for the attribute binding issue. Based on the superior results of IoU, we decided to use this metric for the preservation loss.
>
> ### (d) Spatial Loss and Logistic Sigmoid:
> Our goal is to smoothly guide the generation at each step so that the final image satisfies compositional constraints. To achieve this, our method applies slight adjustments during each generation step, ensuring that each compositional issue is addressed without damaging other aspects. In spatial relationships, the sigmoid keeps gradients bounded even when errors are large, preventing gradient explosions. Moreover, extreme saturation and the corresponding vanishing of gradients do not occur in our setting because the spatial coordinates lie within a bounded range, ensuring that the sigmoid’s input never exceeds the maximum of the image’s width and height, which is 16 in our method (and usually much lower than this upper bound).
>
> ## Inference Time Reporting:
> In Table 5, we compare our method with other baselines in terms of inference time, and a detailed analysis of the results is provided in Section 6.2, under “Inference-time and Diversity Comparison.”
>
> ## Figure 7 and 8 Clarification:
> Thank you for pointing out the need for clearer captions. We have updated the captions in the revised paper. Our intention with these figures is to demonstrate that the adjusted attention maps of our method can accurately identify objects and their locations in the generated images. Specifically, this process treats attention maps as a tool for object detection. This result occurs only when our method successfully generates all entities in the image, and each entity’s corresponding attention map accurately localizes its position.
>
> In Figure 7, we compare our method with other refinement-based approaches, while Figure 8 provides additional examples of our method and is not a comparison. Moreover, we have explained the process of extracting the bounding boxes from attention maps in Appendix D under “Implementation Details.” If the reviewer still feels these figures are unnecessary, we can remove them.
>
> ## Broader Impact Statement:
> Thank you for your suggestion of including a broader impact statement. As our method enhances the control and fidelity of text-to-image generation, it shares the broader societal risks associated with this technology. We have added an impact statement in Section 8 of the revised paper.

---

### Review · Reviewer_uRLx · 2025-10-26

**Summary Of Contributions:**

This paper proposes a methodology to enhance the quality of generated images in the context of text-to-image diffusion models. In particular, Stable Diffusion is considered in this work. The methodology is based on two main components. The first is initial noise refinement to adjust the initial noise sample to be more suitable for the current prompt. The second is iterative refinement to adjust the current noise at each denoising step, in order to align with the prompt. Both refinements are based on interpretable loss functions that operate at the level of the attention maps between prompt tokens and the image plane.

**Strengths**
- The paper is well written and mostly easy to follow.
- The proposed methodology is intuitive and based on interpretable loss functions.
- The computational overhead imposed by the approach is relatively small.
- The methodology does not require training or fine-tuning.

**Weaknesses**
- The statistical relevance of the quantitative results in unclear.
- The methodology requires individuation of entities, attributes and spatial relationships among entities.
- The methodology might struggle in handling complex prompts.
- The applicability and efficacy when considering other denoising image generation methods other than Stable Diffusion is unclear.

**Audience:**

Yes

**Audience Explanation:**

Overall I think the methodology is interesting and part of TMLR's audience would be interested. This considering the relevance of diffusion based models for text to image generation.

**Broader Impact Concerns:**

I don't think the paper needs a Broader Impact Statement.

**Claims And Evidence:**

No

**Claims Explanation:**

While most claims on the efficacy and efficiency of the proposed approach are reflected by the empirical results, I think that the absence of evaluation based on multiple seeds undermines the claims about the performance improvement over other methods.
In particular I would expect to see average scores and their standard deviation across multiple random seeding for the noise. My understanding is that reported results are for a single seed.
The performance gap seems narrow in many cases, hence the necessity to understand whether they are statistically relevant.

Moreover, the main text reports DA-score and Tifa-score as a mean of comparison between approaches. However, the most relevant advantage seems to correspond to the usage of initial noise refinement, which is applied based on poor results on the DA-score itself. I wonder whether this might unfairly bias the results, and I would invite the authors to discuss this point.
Note it is my understanding that DA and Tifa scores are based on a similar approach, hence this might apply to both metrics.
The paper also reports other scores in the appendix, but it seems the results on them are mixed.

**Requested Changes:**

**Major**
- Performance scores should be computed over multiple trials using different random seeds. Tables should report average score and associated standard deviation.
- The meaning of "improvement up to %" should be clarified. Is this the best case scenario across baselines and evaluation categories? If that is the case, I suggest that the authors highlight some more conservative metric, e.g., average improvement (across categories) w.r.t. the best baseline. Either way, it should be clear to the reader.
- The proposed method has hyperparameters in the form of the weight of gradient updates (Eq. 9 and 13), the "fault" threshold (implicit in Eq.10) and the $t_{max}$ in Algorithm 1. The authors should clarify:
	- How were values fore the hyperparameters found?
	- Was hyperparameter search carried out for the proposed methodology and/or baselines?
	- If feasible, provide a sensitivity study with respect to parameter values.
- The method requires a set of entities, attributes, and relationships to be extracted from the prompt. I think this should be discussed in the main text, as it is not a negligible part of the approach. If one needs to do it by hand it is a considerable overhead, while if done by a large language model (as in this paper) its choice can affect the results.
- The explanation of the Initial Noise Refinement should be improved. Currently it is not entirely clear after a first read. In particular, the text explaining how $A^{init}$ and $A^{ref}$ are computed based on "faulty" or "proper" is convoluted. It should be improved, and Algorithm 2 should formally specify how the $A^{init}$ and $A^{ref}$ sets are computed and updated during iterations, so to avoid misinterpretation.
- Table 4 and the related paragraph on Test-time-sampling methods are unclear. They should clarify what is being compared. It is unclear to me what "Best of N" refers to.

**Minor**
- After Eq. 3: "that have minimal overlap" seems a typo, possibly should have been "to have minimal overlap".
- Clarify the meaning of the plus sign superscript in Eq. 4.
- Terminology could be more polished: e.g., it would be more precise to say that a loss "favors" or "penalizes" a behavior rather than "enforces" it.
- In paragraph "Fine-Grained Noise Refinement", you say "At each time-step of refinement". Do you mean "at each iteration"? As initial noise refinement considers only the first denoising step from my understanding, and the concept of "time-step" was previously used to refer to the process of adding or subtracting noise.
- The DA-score could be explained more thoroughly, at least in the appendix. While it is not the only possible choice, it would help understanding how the alignment between prompt and image is computed.
- Clarify why the user study is performed only on 3 baselines.
- The caption of Table 2 uses a slightly different wording than those of Table 1 and 3. It could be uniformed.
- Table numbering should reflect the order they are mentioned in the text.
- References should follow the main body of the paper, rather than starting on a new page.

---

> ### Author Response · Authors · 2025-12-08
> **Response**
>
> # Response Part 1
>
> **Major Requested Changes**
>
> **Performance Scores:**
>
> None of the compared methods report the standard deviation of scores in their original papers. However, we still evaluated our model across 10 seeds and achieved an average DA-Score of 78.3% across all categories in T2I-CompBench, with a standard deviation of 0.5%. As shown in Table 1, the gap between our model and the runner-up, CONFORM (74%), is much larger than the 0.5% standard deviation. Nonetheless, if required, we can rerun all baseline methods across multiple seeds and report their mean and standard deviation for completeness.
>
> In the edited paper, we mention in the Appendix (under Implementation Details) that we have obtained results across multiple seeds.
>
> **Improvement Percentage:**
>
> An improvement of up to X% is reported relative to the baselines in the category where we achieve the largest margin. In the revised paper, the last line of the abstract and introduction has been updated to report the average improvement over the T2I-CompBench benchmark.
>
> Also, we now include an average column for multi-category benchmarks.
>
> **Hyperparameter Clarification and Sensitivity Study:**
>
> In Eq. 9, $\alpha_t$ represents the gradient update step size. Following prior work [1, 2, 3, 4], we set it to decrease linearly from 1.0 (largest step) to 0.5 (smallest step) over 25 generation steps. Similarly, $t_{\text{max}} = 25$ in Algorithm 1, which specifies the number of steps over which the losses are applied, is also chosen based on prior work [1, 2, 3, 4].
> In Eq. 13, $\alpha$ represents the initial step size and is set to 1.0. This ensures that early refinements—particularly the first refinement of the second generation—have a strong impact and modify the initial noise. We set the faulty threshold to 0.5 because the DA-Score for each <image, question> pair returns a value in [0, 1], representing the probability of a ‘Yes’ answer. Values above 0.5 indicate correct, while values below 0.5 indicate faulty ones. We have included a sensitivity analysis of the faulty threshold in Appendix C.4.3 of the revised paper.
>
> [1] X. Guo et al., “InitNo: Boosting text-to-image diffusion models via initial noise optimization,” in CVPR, 2024.
>
> [2] H. Chefer et al., “Attend-and-Excite: Attention-based semantic guidance for text-to-image diffusion models,” SIGGRAPH, 2023.
>
> [3] T. H. Salih Meral et al., “Conform: Contrast is all you need for high-fidelity text-to-image diffusion models,” in CVPR, 2024.
>
> [4] Y. Li et al., “Divide and Bind Your Attention for Improved Generative Semantic Nursing,” in BMVC, 2024
>
> **Prompt Decomposition Discussion:**
>
> This process has been described in Appendix D as “Prompt Decomposition”, but will also include a short description in the Definition section with reference to the Appendix. We use the LLM GPT-4o to extract entities, attributes, and relationships from the prompt. The extraction is performed using a few-shot approach, with the prompt templates shown in Figures 14 and 15. Overall, the extraction task was straightforward for the GPT family, and testing with different models, such as GPT-4o-mini, consistently produced good results. However, if the reviewer considers it necessary, we can try another LLM to further validate the results.
>
> **Initial Noise Refinement Explanation:**
>
> Thank you for your feedback. We will update the algorithm and the method sections to clarify the preservation loss’s explanation.
>
> **Table 4 and Test-time-sampling Methods:**
>
> The 'best of N (BoN)' concept refers to selecting the best outcome from N independent trials. In our work, BoN (with N = 3) means generating three different images using different random seeds (e.g., 1, 2, 3) and choosing the best one according to our evaluator.
> There are two paradigms that are often confused: Test-Time Sampling (TTS) and Refinement. In the paragraph “Distinction from Test-Time Sampling Methods,” we clarified that our approach differs from TTS methods [1], such as BoN. Specifically, TTS methods generate each sample independently and require the full process to be completed for every attempt. In contrast, our refinement-based method builds upon previous generations, creating a dependency between successive outputs. Importantly, in refinement, the random seed is kept fixed and progressively improved, whereas TTS samples a new seed at each iteration. Fixing the seed (compared to BoN) creates a more challenging setting, as some initial seeds may be suboptimal and cannot be trivially corrected through refinement.
>
> [1] N. Ma et al., Inference-time scaling for diffusion models beyond scaling denoising steps, arXiv 2025.

---

> ### Author Response · Authors · 2025-12-08
> **Response**
>
> # Response Part 2
>
> **Minor Requested Changes**
>
> **Typo Correction:**
>
> Thank you for catching the typo. We fixed it in the revised version.
>
> **Clarify Notation:**
>
> Thank you for pointing this out. The plus sign is a shorthand notation for taking the maximum with zero. Concretely, this means that you first compute the elementwise difference between the two attention maps, then take the maximum of each element with zero. The resulting attention map has the same shape as the inputs (16×16), but each element retains only the positive value of the corresponding difference, with negative values replaced by zero. We have updated the equation in the paper.
>
> $$\max\big( A_{e_1}^t - A_{e_2}^t \big)^{+} = \max\big(A_{e_1}^t - A_{e_2}^t, 0\big)$$
>
> **Terminology Refinement:**
>
> We sincerely thank you for your kind suggestions, and we have incorporated the changes in the revised paper.
>
> **Clarify Time-step/Iteration:**
>
> We are referring to the steps of entity refinement when, at each step, we choose a faulty entity and correct it. This is separate from the denoising steps to which we refer as “iterations”. We corrected the text to clarify that this is separate from the denoising steps.
>
> **DA-Score Explanation:**
>
> In Appendix B of our submitted paper, we described the evaluation process using our modified DA-Score. We explained the changes made to adapt the DA-Score to our task and provided a description of our evaluation procedure. Please refer to this section for more information.
>
> **User Study Baselines:**
>
> Based on the results in Tables 1, 2, we selected only the strongest state-of-the-art competitors to our approach for human evaluation. Given the high cost of human assessment, the evaluation was therefore restricted to the proposed method and its most competitive baselines.
>
> **Table Caption Uniformity:**
>
> Thank you for pointing this out, we updated the caption of Table 2 to make it consistent with the others.
>
> **Table Numbering:**
>
> Thank you for bringing this to our attention. We have updated the table order to reflect their sequence in the paper.
>
> **References Placement:**
>
> Thank you for pointing this out. We updated the paper so that the references immediately follow the main body.

---

> > ### Comment · Reviewer_uRLx · 2025-12-22
> >
> > I thank the authors for their response, they addressed most of my concerns.
> >
> > I do think that reporting standard deviation and the number of seeds used (now the paper just says "multiple") is important to compare different models. In this sense, adding this measure across the different experiments, at least for the training free baselines, would strengthen the evidence of performance improvements being statistically significant.
> >
> > Regarding testing other LLMs for prompt decomposition, I think that is not necessary.

---

### Decision · Action_Editor_zTjR · 2025-12-23

**Recommendation:** Reject

**Audience:**

Yes

**Audience Explanation:**

This paper targets improvements in diffusion-based text-to-image generation, a highly active research area, and its findings are likely to be of interest to the TMLR audience.

**Claims And Evidence:**

No

**Claims Explanation:**

This paper proposes a training-free method to improve compositional text-to-image generation in diffusion models. After rebuttal, it received two Leaning Accept and one Leaning Reject. The approach is interesting, effective, and highly practical, as it operates purely at inference time without requiring retraining or fine-tuning.

However, reviewers raised concerns about the lack of ablation studies for key modeling choices, which were not adequately addressed in the rebuttal. Several issues therefore remain unresolved:

1. Functional role of EAR losses. While the authors cite precedent for the terminology, invoking prior misuse does not justify it here, especially for a method explicitly described as “training-free.”

2. Min–max normalization. The motivation for this choice is unclear. Since softmax is shift-invariant, min–max scaling effectively acts as adaptive temperature scaling. Neither the paper nor the cited work (Guo et al., 2024) discusses this, and no ablation is provided to justify the choice.

3. 16×16 resolution constraint. Although common in prior work, no evidence is given that this resolution is optimal. Without ablation, the constraint remains unjustified.

4. Sigmoid saturation. The argument that values are bounded by 16 due to the 16×16 constraint is unconvincing, as sigmoid saturation occurs well within these bounds. Moreover, this explanation is circular, relying on another unablated design choice.

5. Inference time comparisons. While inference times are reported, they are not incorporated into the main performance comparisons, which was the reviewer’s original concern.

Despite improved presentation after rebuttal, the paper still lacks sufficient empirical justification for key modeling choices. The reliance on precedent rather than systematic ablation is a major weakness, especially for a training-free method where such choices are critical. As a result, the work remains insufficiently sound.

**Resubmission Of Major Revision:**

The authors may consider submitting a major revision at a later time.